# Single-housing–induced islet epigenomic changes are related to polymorphisms in diabetic KK mice

Takao Nammo[1,2,5], Nobuaki Funahashi[1,10], Haruhide Udagawa[1,11], Junji Kozawa[2,5], Kenta Nakano[3], Yukiko Shimizu[3], Tadashi Okamura[3], Miho Kawaguchi[1], Takashi Uebanso[1,4], Wataru Nishimura[1,6,7], Masaki Hiramoto[1,8], Iichiro Shimomura[2], Kazuki Yasuda[1,9]

A lack of social relationships is increasingly recognized as a type 2 diabetes (T2D) risk. To investigate the underlying mechanism, we used male KK mice, an inbred strain with spontaneous diabetes. Given the association between living alone and T2D risk in humans, we divided the non-diabetic mice into singly housed (KK-SH) and group-housed control mice. Around the onset of diabetes in KK-SH mice, we compared H3K27ac ChIP-Seq with RNA-Seq using pancreatic islets derived from each experimental group, revealing a positive correlation between single-housing–induced changes in H3K27ac and gene expression levels. In particular, single-housing–induced H3K27ac decreases revealed a significant association with islet cell functions and GWAS loci for T2D and related diseases, with significant enrichment of binding motifs for transcription factors representative of human diabetes. Although these H3K27ac regions were preferentially localized to a polymorphic genomic background, SNVs and indels did not cause sequence disruption of enriched transcription factor motifs in most of these elements. These results suggest alternative roles of genetic variants in environment-dependent epigenomic changes and provide insights into the complex mode of disease inheritance.

## Introduction

Type 2 diabetes (T2D) is a continual clinical concern with an enormous impact on public health and is socioeconomically important (Ogurtsova et al, 2017). Many studies have shown the involvement of both genetic predispositions and environmental factors in the development of T2D (Zheng et al, 2018). Since their initial discoveries (Sladek et al, 2007; Yasuda et al, 2008), genome-wide association studies (GWAS) have identified several hundred susceptibility loci for T2D (Udler et al, 2019). One proposed mechanism of a causal single-nucleotide polymorphism (SNP) is the perturbation of epigenetic regulation through disruption of the transcription factor (TF) binding motifs (Farh et al, 2015). Environmental changes also influence gene expression via epigenetic mechanisms (Unnikrishnan et al, 2017). However, despite the overall involvement of environmental factors, sequence variants, and the chromatin state, there is little in vivo evidence regarding its pathophysiology.

Defects in insulin secretion are central to T2D. Various rodent models have been established as potential surrogates for humans (Kleinert et al, 2018); however, how they can be applied remains unclear. Previously, using simple environmental manipulation, we found that a high-fat diet modified beta-cell function through epigenomic changes in C57BL/6J mice (Nammo et al, 2018). However, the effects of environmental factors on the susceptibility to T2D have not yet been determined.

KK mice belong to an inbred strain originally derived from experimental albino mice obtained from Kasukabe District in the suburbs of Tokyo, Japan (Kondo et al, 1957; Ikeda, 1994). Initially, this strain was established through a selection regime for reduced variance in body weight, gentle nature, and high reproductivity to produce an ideal experimental mouse line (Kondo et al, 1957) but was reported to be hereditarily obese and diabetic several years later (Nakamura, 1962). Although it is regarded as a model of diabetes because of insulin resistance caused by mild obesity (Dulin & Wyse, 1970), defects in the insulin secretory response to glucose and pathological changes in beta cells exist in these mice (Ikeda, 1994). Moreover, similar to human T2D, some diabetic complications

[1]Department of Metabolic Disorder, Diabetes Research Center, Research Institute, National Center for Global Health and Medicine, Tokyo, Japan [2]Department of Metabolic Medicine, Osaka University Graduate School of Medicine, Osaka, Japan [3]Department of Laboratory Animal Medicine, Research Institute, National Center for Global Health and Medicine (NCGM), Tokyo, Japan [4]Department of Preventive Environment and Nutrition, Institute of Biomedical Sciences, Tokushima University Graduate School, Tokushima, Japan [5]Department of Diabetes Care Medicine, Osaka University Graduate School of Medicine, Osaka, Japan [6]Department of Molecular Biology, International University of Health and Welfare School of Medicine, Chiba, Japan [7]Division of Anatomy, Bio-Imaging and Neuro-cell Science, Jichi Medical University, Tochigi, Japan [8]Department of Biochemistry, Tokyo Medical University, Tokyo, Japan [9]Department of Diabetes, Endocrinology and Metabolism, Kyorin University School of Medicine, Tokyo, Japan [10]Department of Life Science and Technology, Tokyo Institute of Technology, Yokohama, Japan [11]Department of Registered Dietitians, Faculty of Health and Nutrition, Bunkyo University, Chigasaki, Japan

Correspondence: kyasuda@ks.kyorin-u.ac.jp, kyasuda@ri.ncgm.go.jp; t-nammo@endmet.med.osaka-u.ac.jp

have been previously described in KK mice (Ikeda, 1994; Berndt et al, 2014). By the beginning of the 2000s, genetic approaches, such as QTL mapping, had identified several susceptibility loci associated with diabetes and related diseases in KK mice, suggesting their polygenic nature (Suto et al, 1998; Shike et al, 2001). Environmental factors, including diet, have been reported to play a role in the development of diabetes (Matsuo et al, 1971). Notably, as one of the environmental factors, the impact of social isolation on early mortality has recently been highlighted (Holt-Lunstad et al, 2010). Among the various measures of social isolation, living alone possesses objective characteristics and has been reported to be a risk factor for T2D in humans (Meisinger et al, 2009; Hilding et al, 2015). Interestingly, a similar situation has been observed in KK mice, where single housing is an alternative environmental manipulation that accelerates the onset and development of diabetes, even when mice are fed a regular chow diet (Matsuo et al, 1971). In addition, KK mice are less likely to develop diabetes if they are not exposed to risk factors (Matsuo et al, 1971), as was the case for humans living in the early 1960s (Ogurtsova et al, 2017). Thus, a thorough understanding of the molecular mechanisms of this mouse model could be the key to addressing the global health problems arising from isolation.

Inbred mouse lines with a T2D-like phenotype present a unique opportunity to investigate genomic functions that have not been fully elucidated, because such extreme experimental conditions would never occur in humans. Here, using high-throughput sequencing technologies, we conducted a genome-wide comparative analysis of the histone modification status of cis-regulatory elements and gene expression in pancreatic islets derived from this mouse model and examined single-housing (SH)–dependent epigenomic changes.

# Results

## Body weight and glucose tolerance in KK mice under different stocking densities

We used KK mice to investigate how environmental risk factors induce pancreatic beta-cell dysfunction. We divided 9-wk-old male KK mice into two groups: singly housed (KK-SH) and group-housed (KK-GH) control mice (Fig 1A). We found that KK-SH mice gained more weight (Fig 1B) and became more hyperglycemic (KK-GH, $P$ = 0.7; KK-SH, $P$ < 0.001; linear trend test; Fig 1C) over time than did the KK-GH mice. This phenomenon was strain-dependent as the same experiment did not reveal a significant difference in body weight (Fig S1A) and non-fasting blood glucose levels (Fig S1B) in C57BL/6J mice. At 11 wk of age, KK-SH mice showed slightly elevated body weight (KK-GH 32.4 ± 1.3 g; KK-SH 35.8 ± 2.6 g, $P$ < 0.005; Fig 1B) and non-fasting blood glucose levels (KK-GH 6.1 ± 1.1 mmol/l; KK-SH 8.8 ± 1.6 mmol/l, $P$ < 0.001; Fig 1C) compared with KK-GH mice with significant hyperinsulinemia (KK-GH 305.2 ± 183.7 pmol/l; KK-SH 1,390.9 ± 1,083.6 pmol/l, $P$ < 0.005; Fig 1D). An intraperitoneal glucose tolerance test revealed glucose intolerance (Fig 1E), with a blunted insulin secretion response (Fig 1F). Thus, 11-wk-old KK-SH mice showed metabolic features that are often observed in the prediabetic state (Borch-Johnsen et al, 2004; Tabák et al, 2012).

## Environmental impact on transcriptomic profile through epigenomic regulation in KK mouse pancreatic islets

Next-generation sequencing was used to characterize the global transcriptome and epigenome of the pancreatic islets (Fig 1A). Because the data aligned to the mm10 genome frequently contained sequence variants that might have influenced the analysis (Fig S2A), a custom KK genome and annotation were prepared by incorporating SNPs (SNVs) and indels of KK/HIJ mice (Yalcin et al, 2012), a closely related subline (Clee & Attie, 2007). Using this method, nucleotide mismatches were markedly reduced (Fig S2B), facilitating strain-level analysis.

RNA sequencing (RNA-Seq) revealed that 492 genes were differentially expressed between the groups at 11 wk of age (Fig 2A; Table S1). Gene ontology (GO) analysis using the Database for Annotation, Visualization, and Integrated Discovery (Huang et al, 2009) revealed that the most up-regulated genes in KK-SH mice were enriched in terms of cell cycle and related mechanisms (false discovery rate [FDR] < 0.01, Fig S3A; Table S2). In contrast, the most down-regulated genes were enriched in distinct functions, including those linked to immune system processes and cell adhesion (Fig S3B; Table S3).

For epigenomic analysis, we used H3K27ac chromatin immunoprecipitation sequencing (ChIP-Seq) (Rada-Iglesias et al, 2011). After alignment, the data were processed for differential enrichment analysis using SICER2 (Fig 1A; Table S4) (Xu et al, 2014). We converted the genomic coordinates of the merged H3K27ac regions into mm10 assembly using UCSC liftOver (Hinrichs et al, 2006) with a custom chain file (Pracana et al, 2017) and identified 46,288 merged H3K27ac regions, consisting of 19,981, 6,055, and 20,252 regions showing an increase, decrease, and no change, respectively (Tables S5 and S6). We also found that these merged H3K27ac regions exhibited distinct genomic distributions (Figs 2B and S4A–D; Table S6); increased H3K27ac showed relative accumulation around the transcription start sites (TSSs) (Figs 2B and S4A), whereas decreased H3K27ac tended to localize distal to the genes (Figs 2B and S4C).

Next, we examined the relationship between the changes in H3K27ac expression and differential gene expression. Using the "single nearest gene" association rule proposed in GREAT (McLean et al, 2010), we empirically defined a median of 49 kb nonredundant long-range cis-regulatory element domains (LCREDs) for each gene (Fig 2C; Table S7), where differential H3K27ac regions should exhibit transcriptional regulation. We found various changes in H3K27ac in LCREDs; ~47.1% (6,668/14,154) of the LCREDs exhibited multiple types (Fig S4E; Table S8). Thus, we selected 249, 2,047, or 3,864 genes associated exclusively with the decreased, unchanged, or increased H3K27ac level, respectively (Fig 2D; Table S8), and found a significant relationship between changes in H3K27ac and gene expression ($P$ = 2.1 × 10$^{-10}$, Kruskal–Wallis test with Dunn's post hoc test; Fig 2E).

## Motif enrichment analysis for differentially enriched H3K27ac in KK mice

H3K27ac ChIP-Seq is useful for identifying cis-regulatory elements that mediate environmental effects (Gosselin et al, 2014; Nammo

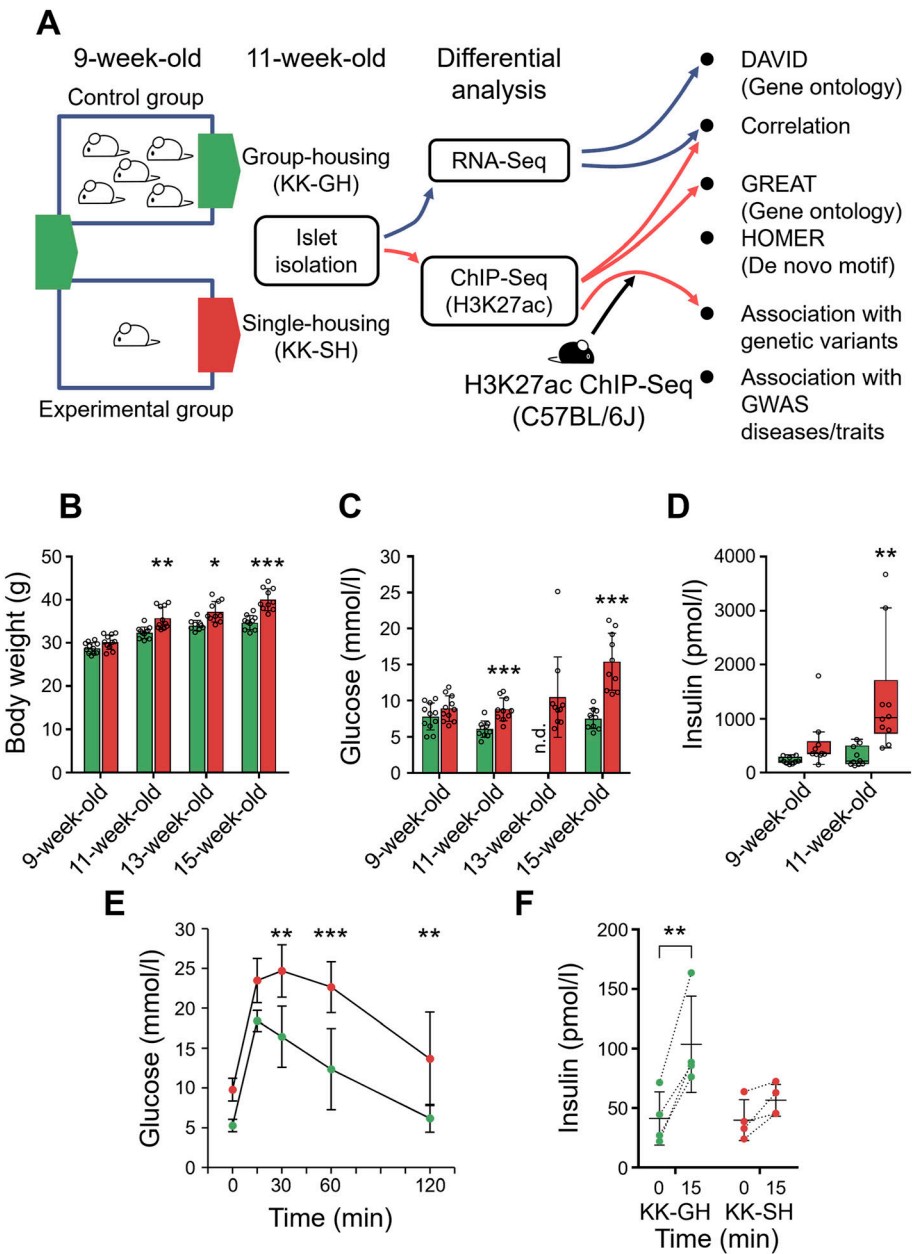

**Figure 1. Single housing drives earlier weight gain and diabetes onset in KK mice.**
**(A)** Schematic diagram of mouse experiments and the analytic flow. **(B, C, D, E, F)** Characterization of the metabolic phenotype of 9- to 15-wk-old mice after grouping mice into those under group-housing (GH) or single-housing (SH) conditions. **(B)** Body weight of KK-GH (green bars) and KK-SH (red bars) mice. KK-GH, n = 11 (9–11 wk old) or 10 (13–15 wk old); KK-SH, n = 11 (9–11 wk old) or 10 (13–15 wk old). **(C)** Non-fasting glycemia of KK-GH (green bars) and KK-SH (red bars) mice. KK-GH, n = 11 (9 wk old) or 10 (11–15 wk old); KK-SH, n = 11 (9 wk old) or 10 (11–15 wk old). The data for mice at 9 wk old were obtained 2 d after grouping. n.d., not done. A linear trend test was performed to determine an age-dependent increase in blood glucose levels for each group. *P*-values were calculated using multiple t tests using the two-stage step-up false discovery rate method of Benjamini, Krieger, and Yekutieli. **(D)** Non-fasting insulin of KK-GH (green boxes) and KK-SH (red boxes) mice. KK-GH, n = 10; KK-SH, n = 10. The data for mice at 9 wk old were obtained 2 d after grouping. **(E)** Intraperitoneal glucose tolerance test (glucose excursion). Green circles, KK-GH, n = 4; red circles, KK-SH, n = 4. **(F)** IPGTT (insulin values). Green circles, KK-GH, n = 4; red circles, KK-SH, n = 4. Mean ± SD in (B, C, E, F). In (D), the horizontal lines inside the boxes are the median values. The upper and lower ends of the boxes are the 75th and 25th percentiles of values, respectively. The vertical lines from the ends of the box (whiskers) represent the smallest or largest observed values within 1.5 times the interquartile range. *P*-values in (B, D, E, F) were calculated using two-way analysis of variance with a Sidak post hoc test. *$P < 0.05$, **$P < 0.01$, ***$P < 0.001$.

et al, 2018). We focused on the differentially enriched regions displaying a uniform direction (increase, decrease, or no change) and performed a de novo motif search using HOMER to identify the binding sequences of the TFs (Heinz et al, 2010). We found 14 significant motifs in 20,295 regions with increased H3K27ac levels (Tables S4 and S5); however, neither matched well with the consensus sequences, as evidenced by the relatively low motif scores (Figs 3A and S5A). Thus, we used HOMER analysis for known motifs, revealing highly significant consensus motifs such as NRF1 ($P = 10^{-30}$) and ZBTB33 ($P = 10^{-24}$) (Fig S6A).

Next, we performed a de novo motif search for the regions with decreased H3K27ac levels (Table S4). Among the six detected motifs,

four were highly similar to the known consensus motifs HNF1B ($P = 10^{-38}$), NEUROG2 ($P = 10^{-32}$), FOXA1 ($P = 10^{-23}$), and RFXDC2 ($P = 10^{-15}$) (Fig 3B), among which *HNF1B* is known to be responsible for maturity-onset diabetes of the young (MODY) type 5 (Horikawa et al, 1997). Furthermore, each motif was similar to that of HNF1A, NEUROD1, FOXA2 (FOX:Ebox), and RFX6 (Fig 3B), which have been established as the causative or susceptible genes of human MODY3 (Yamagata et al, 1996), MODY6 (Malecki et al, 1999), T2D (Gaulton et al, 2015), and neonatal diabetes mellitus (Smith et al, 2010), respectively, suggesting the possible involvement of similar pathological mechanisms of human diabetes in KK-SH mice. In addition, HOMER analysis of unchanged H3K27ac regions

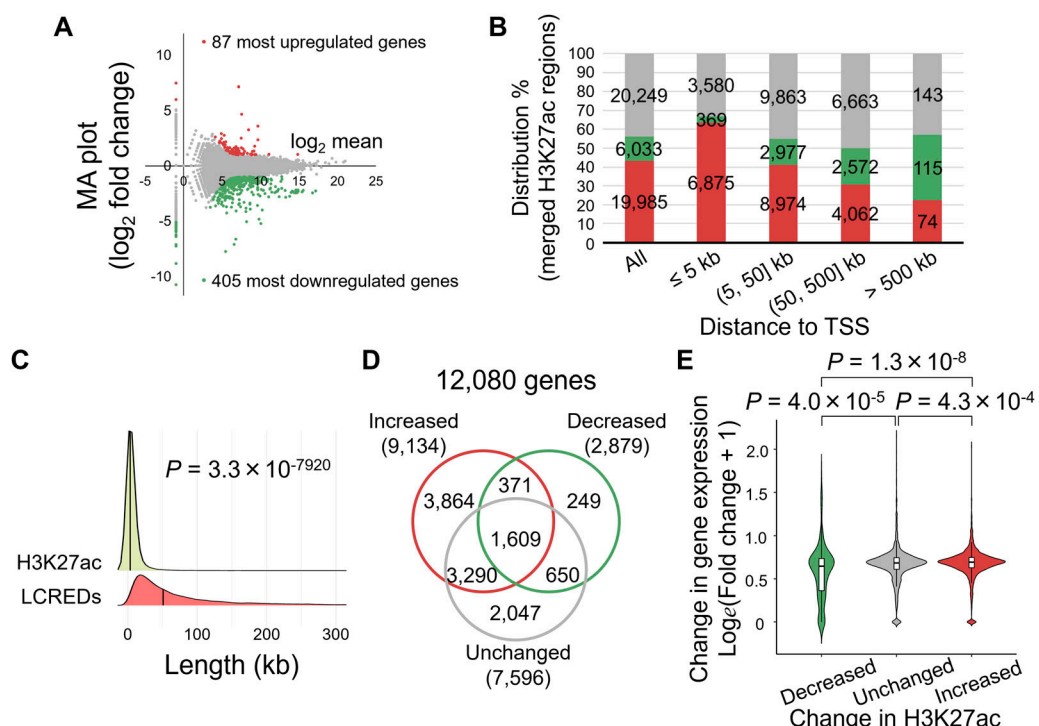

**Figure 2. Overview of differential H3K27ac regions shaping the cis-regulatory landscape in relation to the transcriptome in pancreatic islets derived from group-housed KK and singly housed KK mice.**

**(A)** MA plot of total RNA-sequencing data. The vertical axis indicates the M values that are the $\log_2$ fold change of experimental to control normalized read counts, whereas the horizontal axis indicates the A values that are the intergroup $\log_2$ mean of normalized read counts between the samples. The red and green plots represent top-ranked 87 and 405 genes exhibiting up-regulation and down-regulation, respectively, with a probability threshold ≥0.9 using NOISeq. **(B)** Genome-wide assessment of localization of differentially enriched H3K27ac regions with single-housing–dependent changes in chromatin immunoprecipitation–sequencing signal for the distance of ≤5, 5–50, 50–500, or >500 kb from transcription start sites. Red bars, increased H3K27ac sites; green bars, decreased H3K27ac sites; gray bars, unchanged H3K27ac sites. **(C)** Density plot showing the size distribution of merged H3K27ac regions and long-range cis-regulatory element domains (LCREDs) of genes that were defined using the "single nearest gene" rule proposed in GREAT (http://great.stanford.edu/public/html/). The median values are indicated by the vertical lines. *P*-value was calculated using the Mann–Whitney *U* test. **(D)** Venn diagram showing the number of genes associated with differentially enriched H3K27ac regions that were increased, decreased, or unchanged after single housing. Genes that were incalculable using NOISeq were excluded, although the comprehensive data are shown in Fig S4E. **(E)** Violin plot showing the differential expression of genes associated with cis-regulatory elements, the activity of which was only decreased (n = 243), only unchanged (n = 2,013), or only increased (n = 3,726). The vertical lines from the ends of the boxes (whiskers) represent the smallest or largest observed values within 1.5 times the interquartile range. *P*-values were calculated using the Kruskal–Wallis test followed by Dunn's post hoc test.

(Table S4) revealed that motifs for MEF2 family TFs were the most significant ($P = 10^{-51}$, Figs 3C and S5B). For both decreased and unchanged H3K27ac regions, the enrichment of known motifs (Fig S6B and C) was not as significant as that of de novo motifs (Fig 3B and C).

### Functional enrichment analysis for differentially enriched H3K27ac in KK mice

Given the relationship between a SH-induced decrease in H3K27ac and human diabetes (Fig 3B), we performed a functional analysis using GREAT to gain insights into the various H3K27ac regions (Table S9). Notably, the decreased H3K27ac regions were highly enriched for ontology terms linked to the pancreatic beta-cell function ($P = 3 \times 10^{-36}$; term "abnormal insulin secretion") and insulin action ($P = 2 \times 10^{-26}$; term "increased insulin sensitivity") (Table S9), whereas increased and unchanged H3K27ac regions were not (Table S9). Thus, decreased H3K27ac levels may play a role in the diabetes exhibited by KK-SH mice.

### Sequence disruption of critical TF binding motifs was unprevailing in SH-induced H3K27ac decrease in KK mice

Because SH-induced weight gain and diabetes are a set of phenotypes that are transmissible from parent to child in KK mice, it could be possible to identify the associated changes in the genomic DNA sequences relative to the reference genome of C57BL/6J mice. We first tested whether sequence variants were involved in the decrease in H3K27ac caused by one or more nucleotide changes in critical TF motifs (Fig 3B). After identification of sites for known TF motifs on the basis of the mm10 assembly using a HOMER scan-MotifGenomeWide.pl script, we found that 97.6% (5,910/6,055) of the decreased H3K27ac regions contained at least one motif, including HNF1B, HNF1A, NEUROD1, FOX:Ebox, FOXA2, and RFX6 (Table S10). However, the detection of SNVs or indels revealed that there was depletion of small variants in the merged H3K27ac regions, with only 16.3% (965/5,910) of the decreased H3K27ac regions exhibiting an overlap between the TF motifs and sequence variants (Table S11). We found that, for more than 75% motifs including an SNV,

## A  Increased H3K27ac

| Motif | TFs | Score | P |
|---|---|---|---|
| | EGR1 | 0.686 | $10^{-97}$ |
| | BREu | 0.682 | $10^{-86}$ |
| | ELK1 | 0.728 | $10^{-64}$ |
| | ZNF460 | 0.539 | $10^{-53}$ |
| | ZBTB33 | 0.785 | $10^{-38}$ |
| | MED-1 | 0.677 | $10^{-33}$ |
| | TRP | 0.694 | $10^{-25}$ |
| | ZFP691 | 0.715 | $10^{-21}$ |

## B  Decreased H3K27ac

| Motif | TFs | Score | P |
|---|---|---|---|
| | HNF1B | 0.982 | $10^{-38}$ |
| | HNF1A | 0.970 | $10^{-38}$ |
| | HNF1A (Jaspar) | | |
| | NEUROG2 | 0.982 | $10^{-32}$ |
| | NEUROD1 | 0.980 | $10^{-32}$ |
| | NEUROD1 (GSE30298) | | |
| | FOXA1 | 0.878 | $10^{-23}$ |
| | FOX:Ebox | 0.870 | $10^{-23}$ |
| | FOX:Ebox (GSE47459) | | |
| | PROX1 | 0.623 | $10^{-21}$ |
| | RFXDC2 | 0.896 | $10^{-15}$ |
| | RFX6 | 0.790 | $10^{-15}$ |
| | RFX6 (GSE62844) | | |
| | STAT2 | 0.754 | $10^{-15}$ |

## C  Unchanged H3K27ac

| Motif | TFs | Score | P |
|---|---|---|---|
| | MEF2D | 0.961 | $10^{-51}$ |
| | FOXO1 | 0.927 | $10^{-47}$ |
| | OSR2 | 0.715 | $10^{-42}$ |
| | IRF4 | 0.823 | $10^{-39}$ |
| | ELK1 | 0.587 | $10^{-31}$ |
| | FOXE1 | 0.816 | $10^{-25}$ |
| | DBP | 0.832 | $10^{-21}$ |
| | RFX5 | 0.913 | $10^{-20}$ |

**Figure 3.  De novo motif discovery by HOMER in H3K27ac regions with single-housing–induced change.**
**(A)** Analysis of increased H3K27ac regions. Among 14 motifs identified (Fig S5A), the eight top-ranked motifs are shown. **(B)** Analysis of decreased H3K27ac regions. Six motifs are identified. Motifs for HNF1B, NEUROG2, FOXA1, and RFXDC2 were similar to those for HNF1A, NEUROD1, FOX:Ebox, and RFX6, respectively. **(C)** Analysis of unchanged H3K27ac regions. Among 16 motifs identified (Fig S5B), the eight top-ranked motifs are shown. As a background, the custom reference genome for KK mice was used.

single base pair substitution caused disruption of the binding affinity as estimated by the "No Read Left Behind (NRLB)" energy prediction algorithm (Fig S7; Table S12) (Rastogi et al, 2018) and could indeed be involved in the SH-induced local decrease in H3K27ac levels. However, in most H3K27ac decreases caused by SH, motif disruption is unlikely to be the direct cause. Moreover, we did not find a negative correlation between changes in H3K27ac and sequence variant density (Figs 4A and S8A; Table S6), suggesting that sequence variants inside the H3K27ac regions generally had relatively little effect on the local SH-induced decrease in H3K27ac levels.

### Sequence variant distribution of KK mice was biased toward gene loci exhibiting the SH-induced H3K27ac decrease and mouse-selective presence/absence of H3K27ac

To further investigate the genetic effects on the islet epigenome in KK mice, we referred to our previous H3K27ac ChIP-Seq data from diet-induced obese C57BL/6J mice (Fig 4B) (Nammo et al, 2018), which showed mild hyperglycemia without overt diabetes. Inbred mouse strains are homozygous, and the detection of strain-selective presence or absence of H3K27ac could enable the identification of local regulatory genetic variants using the concept of histone acetylation quantitative trait loci (Sun et al, 2016), whose epigenomic activity is robust irrespective of the environment. Because the islet epigenomic state in older or diet-induced obese C57BL/6J mice could be used as a control for KK mice owing to the sustained beta-cell function (Avrahami et al, 2015; Nammo et al, 2018), we integrated all islet H3K27ac regions in 11-wk-old KK (KK-GH or KK-SH) and 54-wk-old C57BL/6J (fed a chow or high-fat diet) mice

into a single dataset of 48,185 merged intervals. We identified 37,328 and 42,379 merged regions in C57BL/6J and KK mice, respectively, and 5,806 and 10,857 merged H3K27ac regions that were absent and present selectively in KK mice, respectively (Fig 4B; Table S13). We found that these strain-selective H3K27ac regions harbored significantly more SNVs or indels than did the 31,522 regions shared by both strains (Figs 4C and S8B; Table S13). Moreover, SNV density in H3K27ac regions of the KK-selective absence was significantly higher than that of the SH-induced decrease ($P = 3.1 \times 10^{-355}$; KK-selective absence 0.28/kb; SH-induced decrease 0.14/kb; Fig 4D), suggesting a role in the earlier presence/absence of H3K27ac rather than the environment-induced H3K27ac response in KK-SH mice. Using a de novo motif analysis for the H3K27ac regions of the KK-selective absence, we detected the enrichment of TF motifs for FOXM1 ($P = 10^{-43}$) and LHX2 ($P = 10^{-18}$) (Fig 4E), whereas functional analysis using GREAT revealed the enrichment for the term "decreased insulin secretion" ($P = 8 \times 10^{-11}$; Table S14), implying the relevance of the beta-cell response to obesogenic environment (Davis et al, 2010). We found that 89.9% (5,222/5,806) of the H3K27ac regions contained at least one of the known motifs, including FOXM1 and LHX2 (Table S15); however, only 7.6% (397/5,222) of the H3K27ac regions showed an overlap between these TF motifs and sequence variants (Table S16). As described above for the SH-induced H3K27ac decrease, we estimated using NRLB that for more than 74% of motifs including an SNV, single base pair substitutions were disruptive to the binding affinity (Fig S9; Table S12) and thus could cause the predetermined local H3K27ac absence. However, in most cases of the KK-selective H3K27ac absence, sequence disruption of the enriched motifs is unlikely to be a direct cause.

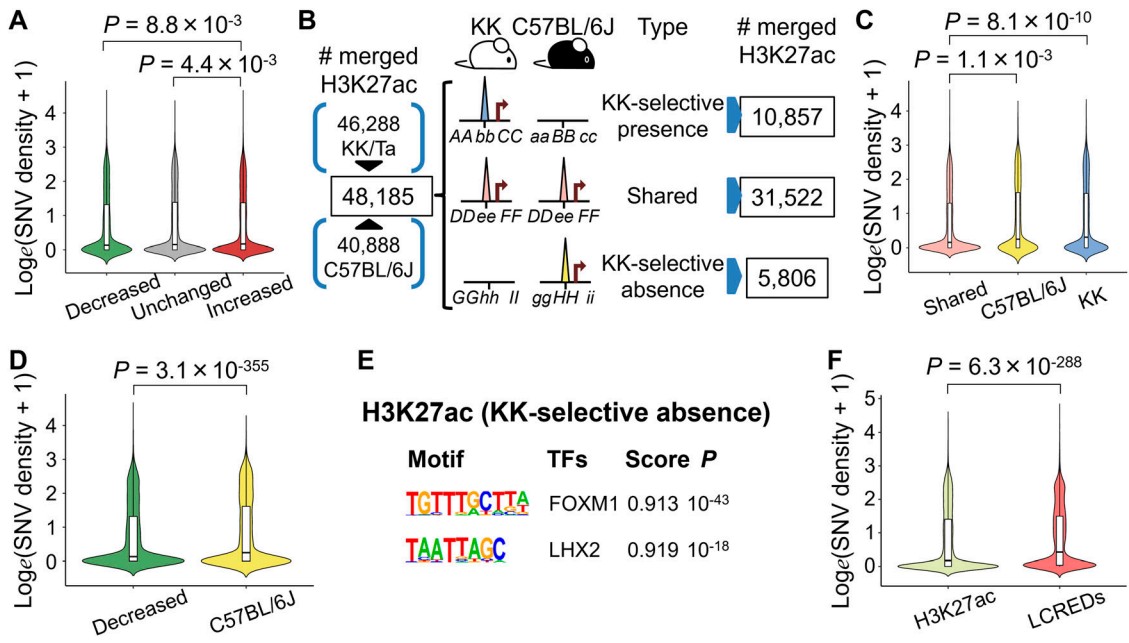

**Figure 4. Density of SNVs in various cis-regulatory elements including strain-selective H3K27ac regions.**
**(A)** Comparison of SNV densities inside H3K27ac regions showing single-housing–dependent change. **(B)** Schematic diagram of analytic flow for the determination of strain-selective H3K27ac regions. The strain-selective presence or absence of H3K27ac was assumed to be dependent on the DNA sequence inside or outside the H3K27ac region. **(C)** SNV densities in H3K27ac regions that are strain-selective or shared between C57BL/6J and KK mice. **(D)** Comparison of SNV densities inside H3K27ac between regions of the single-housing–induced decrease and the C57BL/6J-selective presence. **(E)** De novo motif discovery using HOMER in H3K27ac regions showing the KK-selective absence. Two motifs are identified. **(F)** Comparison of SNV densities in H3K27ac regions and long-range cis-regulatory element domains (LCREDs). Colors of violin plots indicate types of cis-regulatory regions. Green, decreased in singly housed mice (KK-SH); gray, unchanged; red, increased in KK-SH; light pink, shared between C57BL/6J and KK; yellow, C57BL/6J; blue, KK; light green, merged H3K27ac regions between group-housed KK mice and KK-SH; pink, LCREDs. P-values were calculated using the Kruskal–Wallis test followed by Dunn's post hoc test in (A, C), and the Mann–Whitney U test in (D, F).

Because the results obtained thus far did not place much emphasis on the role of small variants in the TF binding motifs in accessible genomic regions, we next examined how SNVs and indels of KK mice were distributed and found that the density of sequence variants was significantly higher outside than inside H3K27ac (Figs 4F and S8C). To further characterize this observation, we used ChromHMM (Ernst & Kellis, 2017) for unsupervised segmentation of the genome. Using an 18-state model derived from the islet ChIP-Seq data of fundamental six kinds of histone modifications, including H3K4me3, H3K27me3, H3K9me3, H3K36me3, H3K27ac (Lu et al, 2018), and H3K4me1 (Wortham et al, 2023) (Figs 5A and S10A), we found that both SNVs and indels were preferentially distributed in the genome segments indicating a relative paucity of these markers (Figs 5B and C and S10A). Therefore, we investigated whether sequence variants outside of H3K27ac had any impact on the various types of H3K27ac changes in the pancreatic islets of KK mice. To test this hypothesis, we examined the aforementioned LCREDs to calculate the density of the included variants (Table S7). Despite the heterogeneity of the H3K27ac regions (Fig 6A), the variants were significantly more frequent in LCREDs harboring at least one KK-selective absence or presence of H3K27ac (Figs 6B–D and S8D and S11A and B). This suggests that the extended genomic landscape of polymorphic loci, as well as variants inside the H3K27ac regions, could provide a mechanistic basis for the strain-selective presence or absence of H3K27ac.

Next, given the relatively small role of sequence variants inside H3K27ac in SH-induced H3K27ac changes as already mentioned, we investigated whether sequence variants outside H3K27ac had any impact on the H3K27ac response to SH. Although they were complicated by various directions of change (Fig 7A), the variant density of KK mice in the mm10 assembly setting was significantly higher in LCREDs where at least one decreased H3K27ac region was included (Figs 7B and S11C; Tables S6 and S7). In contrast, the inclusion of at least one region of increased H3K27ac was associated with a reciprocal decrease in variant density (Figs 7C and S11D; Tables S6 and S7). The negative impact of SNVs was still significant even when the analysis was limited to comparing the three types of LCREDs in which the H3K27ac responses were uniform (Fig 7D), whereas that of indels was obscure (Fig S11E). Taken together, the H3K27ac response to SH is, at least in part, dependent on the genetic background, which is likely to be sequence variants located around and away from the H3K27ac regions.

### The SH-induced decrease and KK mouse-selective absence of H3K27ac cooperatively established an epigenetic architecture in genomic regions for GWAS susceptibility loci of T2D and related diseases/traits

Despite the functional insights obtained from the GREAT analysis, whether SH-induced changes in H3K27ac are associated with specific diseases or traits remains unknown. Given the continuous updates on GWAS signals (Cannon & Mohlke, 2018), we comprehensively identified the genes upstream and downstream of

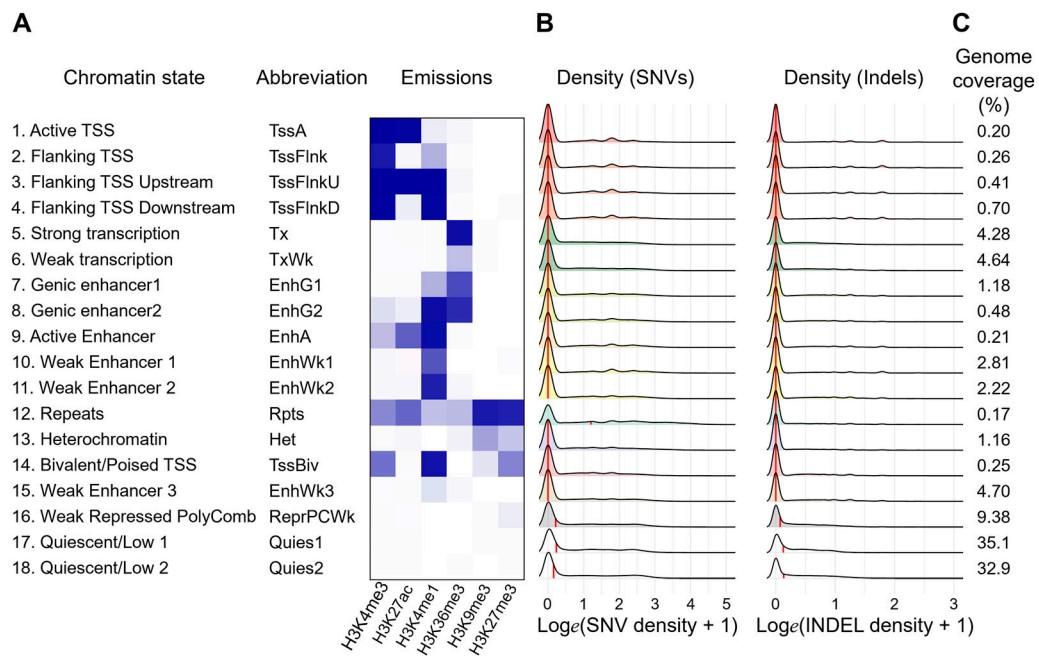

**Figure 5. Annotation of chromatin states in pancreatic islets and distribution of small variants in KK mice.**
**(A)** Definitions and abbreviations of chromatin states and the corresponding histone mark probabilities for the 18-state model in pancreatic islets isolated from C57BL/6J mice. **(B)** Density plot showing the distribution of SNVs (left panel) and indels (right panel) in individual genomic regions with a specified annotation. The median values are indicated by the red vertical lines. **(C)** Average genome coverage of each genomic annotation of the pancreatic islets.

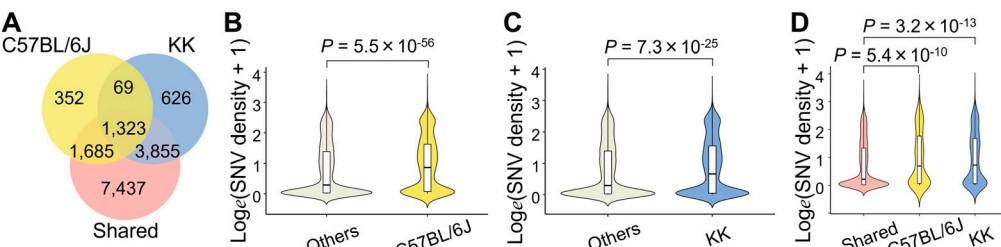

**Figure 6. Strain-selective H3K27ac regions are preferentially found in polymorphic long-range cis-regulatory element domains (LCREDs) of a subset of genes.**
**(A)** Venn diagram showing the number of genes associated with strain-selective and/or shared H3K27ac regions.
**(B)** Comparison of SNV density in the LCREDs including at least one C57BL/6J-selective H3K27ac region with that of others. **(C)** Comparison of SNV density in the LCREDs including at least one KK-selective H3K27ac region with that of others. **(D)** Comparison of SNV density in the LCREDs exclusively including strain-selective or shared H3K27ac regions. $P$-values were calculated using the Mann–Whitney $U$ test in (B, C), and the Kruskal–Wallis test followed by Dunn's post hoc test in (D).

susceptibility variants in the National Human Genome Research Institute (NHGRI)-EBI GWAS Catalog (MacArthur et al, 2017). To minimize inconsistencies because of species differences, we connected only genes with one-to-one orthologs from the Ensembl BioMart resource (Yates et al, 2020) to their LCREDs to define the susceptibility loci in mice. Motivated by the analytical strategy of GREAT (McLean et al, 2010), we used a binomial test to assess the enrichment of susceptible diseases/traits in the H3K27ac regions in both proximal and distal elements, the output results of which included both term-derived and gene-specific enrichment (see the Materials and Methods section).

Regarding SH-induced H3K27ac changes, considerable differences in enriched diseases/traits were identified between the directions (Table S17), suggesting an uneven genomic distribution. We found that 63 of the 144 diseases/traits (43.8%) enriched in decreased H3K27ac were specific to this group, among which two of

the three most significant were fasting blood glucose adjusted for BMI and T2D (Table 1; Table S17). Notably, none of these changes were statistically significant in terms of increased or unchanged H3K27ac levels (Table S17).

We subsequently tested the strain-selective H3K27ac regions with regulatory effects, irrespective of housing conditions. Of the 98 significant diseases/traits enriched in H3K27ac regions showing the KK-selective absence, 77 (78.6%) were specific to this group (Tables 2 and S18). Among the diseases and traits associated with the KK-selective H3K27ac absence, energy expenditure (24 h) was the most significant, followed by T2D.

Although the highest enrichment was observed for both the SH-induced H3K27ac decrease and the KK-selective H3K27ac absence, a large proportion (~30%) of associations with T2D loci were provided by either of the two types of insufficient H3K27ac levels (Fig 7E), suggesting an environmental role in the T2D pandemic

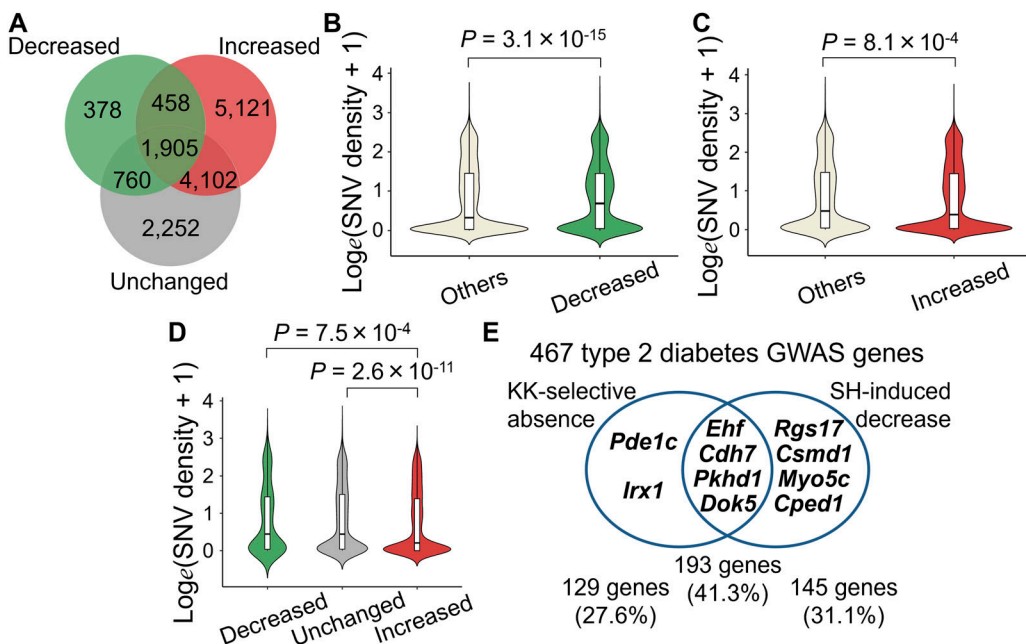

**Figure 7. Decrease in H3K27ac in singly housed KK mice is preferentially found in polymorphic long-range cis-regulatory element domains (LCREDs) of a subset of genes.**
**(A)** Venn diagram showing the number of genes associated with decreased, increased, and/or unchanged H3K27ac regions. **(B)** Comparison of SNV density in the LCREDs including at least one decreased H3K27ac region with that of others. **(C)** Comparison of SNV density in the LCREDs including at least one increased H3K27ac region with that of others. **(D)** Comparison of SNV density in the LCREDs exclusively including decreased, unchanged, or increased H3K27ac regions. **(E)** Venn diagram showing the number of type 2 diabetes susceptibility genes associated with the KK-selective absence and/or the single-housing–induced decrease in H3K27ac. Of all the 467 genes, 10 that showed down-regulated expression after single housing are shown. $P$-values were calculated using the Mann–Whitney $U$ test in (B, C), and the Kruskal–Wallis test followed by Dunn's post hoc test in (D).

**Table 1. Association of decreased H3K27ac regions with various reported genome-wide association study traits and diseases.**

| DISEASE/TRAIT | $P$-value | q value |
|---|---|---|
| Fasting blood glucose adjusted for BMI | $9.0 \times 10^{-13}$ | $1.1 \times 10^{-9}$ |
| Vaginal microbiome MetaCyc pathway (PWY-5121\|superpathway of geranylgeranyl diphosphate biosynthesis II [via MEP]) | $1.3 \times 10^{-11}$ | $1.3 \times 10^{-8}$ |
| Type 2 diabetes | $8.3 \times 10^{-11}$ | $7.0 \times 10^{-8}$ |
| Vaginal microbiome MetaCyc pathway (PWY-922\|mevalonate pathway I) | $1.6 \times 10^{-10}$ | $1.0 \times 10^{-7}$ |
| Fasting blood proinsulin levels | $5.6 \times 10^{-10}$ | $2.6 \times 10^{-7}$ |

Top 5 diseases/traits enriched specifically in decreased H3K27ac.

**Table 2. Association of H3K27ac regions showing the KK-selective absence with various reported genome-wide association study traits and diseases.**

| DISEASE/TRAIT | $P$-value | q value |
|---|---|---|
| Energy expenditure (24 h) | $6.4 \times 10^{-20}$ | $3.2 \times 10^{-16}$ |
| Type 2 diabetes | $1.2 \times 10^{-12}$ | $2.9 \times 10^{-9}$ |
| Adult asthma | $3.3 \times 10^{-10}$ | $5.5 \times 10^{-7}$ |
| Non-alcoholic fatty liver disease histology (lobular) | $2.1 \times 10^{-9}$ | $2.7 \times 10^{-6}$ |
| Obstructive sleep apnea trait (average respiratory event duration) | $4.2 \times 10^{-9}$ | $4.3 \times 10^{-6}$ |

Top 5 diseases/traits enriched specifically in the KK-selective H3K27ac absence.

(Ogurtsova et al, 2017; Zheng et al, 2018). Of the 467 T2D GWAS genes harboring insufficient H3K27ac, 10 genes were found to be significantly down-regulated in the pancreatic islets of KK-SH mice, which showed different patterns of association with either or both types of insufficient H3K27ac levels (Fig 7E; Table S1). These results demonstrate how long-range genetic variations influence environment-dependent and environment-independent epigenomic alterations, which could be at least one of the complex mechanisms underlying the transition from the presymptomatic to symptomatic phases of diabetes in KK mice.

# Discussion

Living alone has gained increasing attention as a risk factor for T2D, and lifestyle factors, including physical activity, diet, and chronic stress, have been hypothesized to be possible mediators; however, the underlying mechanism has not been clarified (Schram et al, 2021). In the present study, we identified cis-regulatory elements sensitive to stocking density in male KK mice and strain-selective cis-regulatory regions by integrating data from C57BL/6J mice (Nammo et al, 2018).

Our analyses allowed the discrimination of at least two types of insufficient H3K27ac statuses in KK-SH mice: KK-selective absence and SH-induced decrease. Although both types often coexist in identical LCREDs with polymorphic genetic backgrounds, the KK-selective absence is distinctive because the element contains relatively abundant sequence variants. This finding is in agreement with previous data (Gaulton et al, 2010; Farh et al, 2015) in which lead or causal GWAS SNPs induced an allelic imbalance of local TF binding or chromatin structure within an identical cellular context, possibly contributing to disease predisposition even in group-housing conditions. Because epigenetic regulation is generally thought to be independent of DNA sequence changes (Holliday, 1987), the association of small variants with environment-dependent epigenomic changes, as suggested by our findings, may be a novel concept in genomic regulation and heritability.

Although the association with T2D was nearly the highest among all 5,054 GWAS diseases/traits for both types of insufficient H3K27ac, the overall significance profiles did not match. The SH-induced H3K27ac decrease exhibited reasonable enrichment for fasting blood glucose adjusted for BMI and T2D, followed by some critical traits relevant to insulin and glucose homeostasis, including fasting blood proinsulin levels. Some well-known complications of T2D were also significantly enriched, including myocardial infarction (early onset) and chronic kidney disease (Table S17).

In the H3K27ac regions showing the KK-selective absence, energy expenditure (24 h) was the most significant factor, followed by T2D. The clinical features often associated with T2D were not as significant, including non-alcoholic fatty liver disease histology (lobular) and obstructive sleep apnea (average respiratory event duration) (Tables 2 and S18). Using a scatter plot matrix of rank correlation testing, the significance levels of all the diseases/traits between sets of H3K27ac regions and the overall enrichment profiles for both types of insufficient H3K27ac were divergent, with a correlation coefficient of 0.446 (Fig S12), despite T2D being the most representative.

This suggests that diabetes is a major disease in KK-SH mice, with a lower impact on other diseases and traits (Ikeda, 1994). Moreover, the SH-induced decrease in H3K27ac levels contributed to an increase in the number of genes with down-regulated expression (Fig 7E). As SNVs and indels were most enriched in large quiescent or repressed chromatin fractions in LCREDs (Fig 5B and C), their regulatory effects on gene expression could also influence tissues other than the pancreatic islets, possibly leading to pleiotropy (Watanabe et al, 2019). Thus, in addition to the already disappeared H3K27ac, environmental factors could enhance the negative epigenomic effects in various LCREDs, where we often observed a multiplicity of susceptibility loci for spontaneous disease progression, including intake of sweets, energy expenditure, and T2D (Tables S17 and S18). Existing studies suggest that only a few out of 10 candidate genes identified using pancreatic islets (Fig 7E), including CSMD1 (Steen et al, 2013) and DOK5 (Cai et al, 2003), are involved in the mechanisms of glucose homeostasis and insulin signaling at the molecular level, respectively, suggesting that many challenges remain to be addressed in future research. Moreover, analyzing tissues other than pancreatic islets may enable the identification of differentially expressed genes that can be used as novel therapeutic targets for systemic treatment.

Although our strategy was effective, it had several limitations. First, the observational time window was limited to the period around the onset of diabetes. A more longitudinal design would complement our findings regarding the unknown epigenetic mechanisms. Second, we did not use chromatin interaction analysis to determine LCREDs (Fullwood & Ruan, 2009; Miguel-Escalada et al, 2019), possibly resulting in less significant conclusions. Third, the TF motifs identified using a single position weight matrix model have low complexity and may not be very predictive of true binding. Fourth, a complete list of human and mouse orthologs is unavailable. Therefore, some of the excluded genes may have been critical. Fifth, because the causal effector genes at each T2D GWAS locus have not been determined yet, we have to say that our approach of taking the disease annotation, principally based on nearest genes, is an approximation. Furthermore, the number of susceptibility loci was small for some GWAS diseases/traits, which possibly resulted in false-negative results. Sixth, this study used bulk tissue of the pancreatic islets. However, the advantages of using pooled tissues from multiple individuals have also been reported (Tehranchi et al, 2016). Seventh, we did not investigate mutations in coding regions. Further research is required to test this hypothesis. Finally, the same experiments were not performed using other T2D models, thus possibly missing other important epigenetic data.

With the great progress in various research fields, including clinical medicine, genetics, and molecular biology, we are experiencing an unprecedented wave of scientific discoveries. However, relatively little is known about how information can be integrated into a new understanding of pathophysiology. To the best of our knowledge, this study is the first to show that epigenomic and transcriptomic profiling of tissues under environmental exposure can be conducted not only to organize overall research findings but also to gain insights into the novel genomic functions underlying the complex patterns of inheritance, thereby providing a promising approach in the field of T2D.

# Materials and Methods

## Mice

Male KK/Ta mice aged 9 wk were purchased from CLEA Japan and randomly assigned to either a group-housing (five mice per cage sized 33.8 × 22.5 × 14.0 cm for length × width × height) or an SH condition in the same-sized cage. They were maintained in a specific pathogen-free facility at a temperature of 22 ± 2°C under a 12-h light/12-h dark cycle, given free access to water, and fed ad libitum on a standard chow (CE-2; 4.6% fat, 69.9% carbohydrate, 25.5% protein, 3.39 kcal/g; CLEA Japan). Body weight measurements and an intraperitoneal glucose tolerance test were performed as previously described (Nishimura et al, 2013). Blood was collected via tail vein bleeding, and glucose levels were measured using Glutest Ace R (Sanwa Kagaku KK). Plasma insulin levels were measured using Mouse Insulin ELISA Kit (Morinaga Institute of Biological Science, Inc.). All animal procedures were approved by the Institutional Animal Care and Use Committee of the National Center for Global Health and Medicine (approval no.18043).

## Isolation of pancreatic islets

Pancreatic islets were isolated at 11 wk of age from both the experimental (single-housed) and control (group-housed) groups using Liberase (Liberase TL; Roche) digestion, as previously described (Nammo et al, 2018). Before islet isolation, non-fasting mice were sacrificed by cervical dislocation after deep anesthesia with sevoflurane.

## Preparation of custom genome sequence of KK mice

To obtain the mouse reference genome of UCSC mm10, we downloaded the FASTA-formatted sequence and a gene annotation GTF file for RefSeq 35,119 transcripts and 25,770 genes (O'Leary et al, 2016) from the iGenome FTP site (available at https://jp.support.illumina.com/sequencing/sequencing_software/igenome.html). We also downloaded VCF-formatted files of SNPs (SNVs) and indels of KK-HIJ mice (available at https://ftp.ebi.ac.uk/pub/databases/mousegenomes/REL-1505-SNPs_Indels/strain_specific_vcfs/). As a reference for KK mice, we generated a consensus sequence by integrating KK/HIJ variants comprising SNVs and indels with the mm10 reference genome using vcftools (vcf-consensus) (Danecek et al, 2011). Furthermore, to convert the genomic coordinates between KK and C57BL/6J mice, flo (Pracana et al, 2017) was used to create chain files using the UCSC liftOver tool (https://genome.sph.umich.edu/wiki/LiftOver) (Hinrichs et al, 2006). The chain file was also used to lift a GTF file of mm10 to generate a custom annotation GTF file of the KK mouse genome using CrossMap (Zhao et al, 2014). We removed unlocalized sequences (suffixed with _random) and unplaced sequences (prefixed with ChrUn_) and used only the assembled chromosomes (chr1-chr19, chrX, chrY, and chrM) for analysis, resulting in a reduction in the number of transcripts from 35,119 to 35,034. Because UCSC GRCm38/mm10 currently represents Ensembl/Gencode release M23 (https://www.gencodegenes.org/mouse/releases.html) (Harrow et al, 2006), we attempted to match each RefSeq mm10 transcript to an Ensembl

mouse gene ID (Yates et al, 2020) by referring to annotation databases, including gene2ensembl (https://ftp.ncbi.nih.gov/gene/DATA/), ncbiRefSeq (https://genome.ucsc.edu/cgi-bin/hgTables), and the mouse genome Gencode release M23, and 34,734 of 35,034 mm10 transcripts were found to be associated with gene identification in the Ensembl database. In total, 24,156 genes were included in the custom GTF file of the KK mouse genome.

## High-throughput RNA sequencing, analyzing differential gene expression and functional annotation

Total RNA was extracted from pancreatic islets, and RNA-Seq was performed as previously described (Nammo et al, 2018). Using Agilent 2100 Bioanalyzer, we determined that the RNA integrity number scores were 9.1 and 8.7 for experimental (singly housed) and control (group-housed) samples, respectively. RNA-Seq library synthesis was performed using 500 ng of the starting material and TruSeq Stranded Total RNA Sample Prep Kit (Illumina). The Ribo-Zero Gold technology was used to remove ribosomal RNA, and RNA fragmentation was activated by heat and cation treatments. Libraries were sequenced using HiSeq 2000 with paired-end reads of 2 × 100 bp for high-throughput total RNA-Seq.

Paired-end RNA-Seq reads were mapped to the UCSC mouse genome GRCm38/mm10 or the custom reference genome for KK mice using HISAT2 version 2.1.0 and StringTie (Pertea et al, 2016), revealing that properly paired reads ranged from 89.4% to 91.0%. Count expression values were estimated using Ballgown and prepDE.py3 for 24,156 genes based on custom annotations for KK mice. Differential gene expression was analyzed using the R/Bioconductor package NOISeq (https://www.bioconductor.org/) (Tarazona et al, 2015). We also used DESeq2 (Love et al, 2014) to obtain normalized counts for the estimation of fold change, and generated an MA plot to visualize differential gene expression between the experimental (singly housed) and control (group-housed) groups. For the functional assessment of genes showing extreme levels of change with a 90% probability threshold assessed by NOISeq, we used the Database for Annotation, Visualization, and Integrated Discovery 6.8 (https://david.ncifcrf.gov/, accessed on 20/09/2021) (Huang et al, 2009).

## Chromatin immunoprecipitation, high-throughput sequencing, and analysis of differential sequencing signals

ChIP was performed as described previously (Nammo et al, 2018). We used a pooling-based method for high-throughput analyses, which proved useful for lowering interindividual variability, according to a previous study using human samples (Tehranchi et al, 2016). For each experimental group, one ChIP sample was prepared by pooling the pancreatic islets derived from 10 animals. After crosslinking in 1% formaldehyde (Merck Millipore), samples were sonicated using Branson Sonifier 450D (Branson Ultrasonics Corporation), followed by incubation with rabbit polyclonal antibodies raised against histone H3K27ac (Active Motif 39135) for immunoprecipitation. Ten nanograms of fragmented ChIP samples was used for ChIP-Seq library synthesis using the NEBNext ChIP-Seq Library Prep Master Mix Set for Illumina (New England BioLabs). Single-end sequencing of the library was performed using the Illumina HiSeq 2000 platform to obtain 1 × 100 bp reads.

To uncover the epigenomic characteristics of KK mice, we integrated previously published ChIP-Seq data from C57BL/6J mice fed a high-fat diet for 27 wk with those from chow-fed controls (Nammo et al, 2018). Sequence reads were aligned to the custom reference genome for KK mice using Bowtie2-2.3.5.1 (Langmead & Salzberg, 2012). SICER2 (https://zanglab.github.io/SICER2/) (Xu et al, 2014) was used to identify differentially enriched H3K27ac regions between the control and experimental ChIP-Seq data without removing duplicate reads (FDR < 0.01). This revealed that 47,004 (20,295 increased, 20,558 unchanged, and 6,151 decreased) and 41,474 (14,201 increased, 22,602 unchanged, and 4,671 decreased) merged with H3K27ac regions in KK and C57BL/6J mice, respectively. Using the UCSC liftOver tool (https://genome.sph.umich.edu/wiki/LiftOver) to convert the genomic coordinates from KK to C57BL/6J mice using the chain file, we obtained 46,288 (19,981 increased, 20,252 unchanged, and 6,055 decreased) and 40,888 (14,012 increased, 22,279 unchanged, and 4,597 decreased) merged H3K27ac regions in the KK and C57BL/6J mice, respectively, for the mm10 reference genome (Fig 4B). These data were visualized using the Integrative Genomics Viewer browser 2.5.2 (http://software.broadinstitute.org/software/igv/) (Robinson et al, 2011) after loading a reference mouse genome for either the KK mice or the GRCm38/mm10 assembly.

To investigate the functional properties of differential H3K27ac regions, we used GREAT (McLean et al, 2010), a web-based functional annotation tool for cis-regulatory elements, focusing on terms belonging to the Mouse Genome Informatics Phenotype (Shaw, 2016) using the default settings for the "single nearest gene" rule. De novo motif enrichment for the differential H3K27ac regions was also assessed relative to the custom background sequences of KK mice using HOMER (Heinz et al, 2010) with the command "findMotifsGenome.pl" with options -size given -mask -mset vertebrates, which was also useful for the identification of the enrichment of known canonical TF motifs especially in the DNA sequences with less degree of freedom (http://homer.ucsd.edu/homer/introduction/basics.html).

As the presence of epigenetic marks can vary depending on environmental conditions, combining our H3K27ac data may provide insights into genomic regions with strain-selective properties. We merged all H3K27ac regions of C57BL/6J and KK mice to identify 48,185 merged intervals in the mm10 reference genome, of which 5,806 or 10,857 were found to be C57BL/6J- and KK-selective, respectively (Fig 4B).

### Estimation of TF-DNA binding affinity using the NRLB tool

To quantitatively estimate the binding affinity of various TF motifs with the associated TFs, we used an analytical tool named NRLB (Rastogi et al, 2018). We downloaded publicly available HT-SELEX datasets for TFs including HNF1A, HNF1B, LHX2 (Jolma et al, 2013; Yang et al, 2017), FOXA2, NEUROD1 (Yin et al, 2017), RFX6, and FOXM1 (Yan et al, 2021). After adapter sequence removal, followed by trimming reads to leave the central 30 base pairs as needed, model fitting was performed using parameters such as the number of rounds and binding mode, as indicated (Table S12). A fastq file of random sequences of the same length was generated (https://github.com/johanzi/fastq_generator) and used as a control when round 0 HT-SELEX data were not available. A mononucleotide

model was generally used, except for the NEUROD1 HT-SELEX dataset, which was trained using a dinucleotide model. According to the authors' tutorial (https://github.com/BussemakerLab/NRLB), we generated an energy logo motif and subsequently scored the binding affinity of all observed DNA sequences of HOMER known motifs and their variants including various base pair substitutions.

### Correlating differential H3K27ac signals with differential gene expression, sequence variant densities, and susceptible genomic loci for various GWAS diseases/traits

To gain further insights into the functions of various H3K27ac regions, such as transcriptional regulation and enrichment analysis of human diseases, it is necessary to associate H3K27ac regions with their target genes. We took advantage of the concept of GREAT (McLean et al, 2010) in terms of gene assignment and used only a binomial test, although the GREAT output, by default, comprised only enriched terms significant by both binomial and hypergeometric tests to avoid false-positive results arising from gene-specific enrichment of proximal and distal elements rather than from enrichment involving multiple genes associated with a certain term. We selected all results that were significant only for the binomial test, because we were interested in covering both types of results with term-originated and gene-specific enrichment.

Because the three optional association rules were reported to work out similarly owing to the inclusion of genomic regions distal to genes, we selected the most simple "single nearest gene" rule so that the defined LCREDs would not overlap with each other. In this setting, each H3K27ac element was assigned to only one target gene as far as this element was located within an LCRED, which was defined as a domain with a maximum of 1 Mb extension each in both the 5′ and 3′ directions to the center point between the gene's TSS and the nearest gene's TSS (http://great.stanford.edu/public/html/). We prepared a bed file specifying all LCREDs, as described below, because it was unavailable. After transferring a bed file of all 142,351 TSSs registered in Gencode release M23 (GRCm38.p6) to GREAT v4.0.4 (http://great.stanford.edu/public/html/, accessed on 26 November 2022), the output data revealed that 25,077 Gencode gene identifications were involved in the GREAT v4.0.4 web platform. We also downloaded a tab-separated values file of the "set of genes for GREAT 4.0" (https://great-help.atlassian.net/wiki/download/attachments/655445/GREATv4.genes.mm10.tsv?version=1&modificationDate=1627412651079&cacheVersion=1&api=v2, accessed on 26/11/2022) and found that 11 GREAT v4.0 genes were not included in the database of Gencode release M23. We then filtered all of these 25,088 results to 21,424 TSSs of non-redundant genes based on Ensembl gene identification, Ensembl gene name, and GREAT gene name, thereby enabling the determination of 21,424 LCREDs using the "single nearest gene" rule.

Next, we assessed the relationship between differential H3K27ac signaling and differential gene expression in the pancreatic islets of KK-GH and KK-SH mice. According to the "single nearest gene" rule, we found that 46,243 of 46,288 merged H3K27ac regions were assigned to 14,162 genes, among which the fold change of DESeq2-normalized read counts could be calculated for 12,076 genes with RNA-Seq reads aligned to the KK-GH sample greater than zero. Because the directions of H3K27ac changes are usually diverse in LCREDs,

we compared the fold changes of 249, 2,047, and 3,864 genes associated exclusively with decreased, unchanged, and increased H3K27ac, respectively. In addition, we intersected various H3K27ac regions and LCREDs with VCF-formatted data on genetic variations in KK mice using BEDTools (Quinlan & Hall, 2010) to estimate variant density as the number of SNVs or indels per 1,000 base pairs.

For the enrichment analysis of human diseases and traits in the collection of certain H3K27ac regions, we downloaded a spreadsheet representing susceptibility variants from the NHGRI Catalog of Published GWAS (gwas_catalog_v1.0.2-associations_e108_r2022-12-21, available at https://www.ebi.ac.uk/gwas/) (MacArthur et al, 2017). Because of the repeated renewal of GWAS signals at various susceptibility loci with a growing number of GWAS (Cannon & Mohlke, 2018), we extensively identified genes upstream and downstream of the susceptibility variants listed in the catalog. To match the gene population between humans and mice, we selected 17,094 one-to-one orthologous genes encoded on assembled chromosomes that were retrieved from the Ensembl BioMart resource (Ensembl Genes 108, available at https://www.ensembl.org/biomart/martview/74d0795cec79ae328b1c02a65f4aa48c, accessed on 29 December 2022) (Yates et al, 2020), from which we finally chose 15,909 genes that were also included in the 21,424 GREAT genes. We assigned an LCRED to each and allowed the sum of all 15,909 regions in the base pairs to be the size of the genome.

Although we initially identified human 18,372 diseases/traits in the GWAS catalog, this number was reduced to 16,459 after gene filtering. Among these, 5,054 diseases/traits with five or more susceptibility genes were selected to make downstream analysis more efficient. To calculate the binomial $P$-value of a GWAS disease/trait for our H3K27ac ChIP-Seq data in mice, we estimated a portion of the genome as the sum of the LCREDs of the susceptibility genes and used the midpoint of the interval to represent each H3K27ac region. For multiple comparison testing, we calculated the FDR using the Benjamini and Hochberg method and used an FDR q value < 0.01 as the threshold for determining the significance.

### Characterization of chromatin states of mouse pancreatic islets using ChromHMM

We performed unbiased genome segmentation using ChromHMM (Ernst & Kellis, 2017) to annotate the chromatin states in mouse pancreatic islets. Because our analysis focused on H3K27ac, we trained the extended 18-state model, instead of the "core" 15-state model (Roadmap Epigenomics Consortium et al, 2015) without H3K27ac, by integrating ChIP-Seq data of histone modifications, including H3K4me3, H3K27me3, H3K9me3, H3K36me3, H3K27ac (Lu et al, 2018), and H3K4me1 (Wortham et al, 2023). The alignment of single- or paired-end sequencing data to the reference mm10 genome was performed using Bowtie2-2.3.5.1 by the default settings, whereas an option "-N 1" was added to allow a mismatch in a seed to increase sensitivity for H3K27me3 and H3K9me3 data. After generation of binarized data using a command "java -mx1200M -jar ChromHMM.jar BinarizeBed -b 200," the training of data was performed through 740 iterations using a command "java -mx1200M -jar ChromHMM.jar LearnModel." Given the variations in epigenetic profiles of enhancers and bivalent and quiescent states among tissues, as reported previously (Velde et al, 2021), we provided annotations and abbreviations

to individual genomic segments as follows: (1) active TSS (TssA), (2) flanking TSS (TssFlnk), (3) flanking TSS upstream (TssFlnkU), (4) flanking TSS downstream (TssFlnkD), (5) strong transcription (Tx), (6) weak transcription (TxWk), (7) genic enhancer 1 (EnhG1), (8) genic enhancer 2 (EnhG2), (9) active enhancer (EnhA), (10) weak enhancer 1 (EnhWk1), (11) weak enhancer 2 (EnhWk2), (12) repeats (Rpts), (13) heterochromatin (Het), (14) bivalent/poised TSS (TssBiv), (15) weak enhancer 3 (EnhWk3), (16) weak repressed polycomb (ReprPCWk), (17) quiescent/low 1 (Quies1), and (18) quiescent/low 2 (Quies2). Among our annotations, the 12th segment showed enrichment both for H3K9me3 and for H3K27me3, and for the other active markers including H3K27ac, which may correspond to the previously deprecated cluster named as "Repeats (Rpts)" because the mapping artifacts to repetitive elements was suspected (Roadmap Epigenomics Consortium et al, 2015). By integrating the annotations of RepeatMasker obtained from the UCSC Genome Browser (Smit AFA, Hubley R, Green P., RepeatMasker Open-3.0. http://www.repeatmasker.org. 1996–2010.), we found that these genomic regions were divided into two groups: one with almost 100% coverage of repeat elements and the other with almost no repeat sequences. The use of paired-end data in our analysis helped reduce the possibility of artifacts. Moreover, an example of H3K9me3 domains showing enrichment of H3K27ac, H3K4me1, and H3K36me3 has been reported (Thorn et al, 2022), implying its relevance, at least in part. We found that these regions occupied the smallest part (0.17%) of the genome and thus had little impact on this study, and we tentatively named them repeats (Rpts). In addition, our results did not include genomic segments corresponding to those initially referred to "Bivalent Enhancer (EnhBiv)." Based on these considerations, we analyzed the statistical differences in the densities of SNVs or indels in these genomic segments.

### Statistical analysis

Statistical analysis for each figure is described in the figure legends. We performed two-sided unpaired $t$ test or two-way analysis of variance using R 4.1.0 (https://www.r-project.org/) or GraphPad Prism 7.04 (GraphPad Software). Non-parametric tests (Mann–Whitney $U$ tests or Kruskal–Wallis tests) were employed when the data did not follow a normal distribution. Error bars represent SD.

### Accession numbers

Data from this study have been deposited in the DNA Databank of Japan (DDBJ) under the accession numbers DRA015613 (RNA-Seq) and DRA015614 (ChIP-Seq). Our previous ChIP-Seq data from C57BL/6J mice were deposited in ArrayExpress under the accession number E-MTAB-6719 (Nammo et al, 2018).

# Data Availability

The next-generation sequencing data generated in this study were submitted to DDBJ (https://www.ddbj.nig.ac.jp/ddbj/index-e.html) under the accession numbers DRA015613 (RNA-Seq) and DRA015614 (ChIP-Seq). The custom genome sequence of KK mice in the present study and a custom annotation GTF file and chain file for use in

the UCSC liftOver tool were submitted to the Figshare database (10.6084/m9.figshare.21995645): Filename—genome.KK-HiJ.fa (KK mouse genome) and KK-HiJ.genes_annotation.gtf (annotation GTF file), liftover_KKHiJToC57BL6J.chn (a chain file for the conversion of genomic coordinates from KK to mm10).

## Supplementary Information

## Acknowledgements

We thank Mr. D Suzuki, Ms. H Shiina, Ms. T Shibuya, Ms. K Nagase, and Ms. N Ishibashi for their assistance. We would like to thank Editage (www.editage.jp) for English language editing. This work was supported by the Japan Society for the Promotion of Science (JSPS) Grants-in-Aid for Scientific Research (KAKENHI) (21K08577), National Center for Global Health and Medicine (29-1032) (to T Nammo), JSPS KAKENHI (16K01850), and National Center for Global Health and Medicine (28–1201) (to K Yasuda).

### Author Contributions

T Nammo: conceptualization, resources, data curation, software, formal analysis, funding acquisition, validation, investigation, visualization, methodology, project administration, and writing—original draft, review, and editing.
N Funahashi: formal analysis, validation, and investigation.
H Udagawa: formal analysis, validation, investigation, methodology, and writing—original draft, review, and editing.
J Kozawa: resources, validation, investigation, and writing—original draft, review, and editing.
K Nakano: resources, validation, and investigation.
Y Shimizu: resources, validation, and investigation.
T Okamura: resources, validation, and investigation.
M Kawaguchi: validation, investigation, methodology, and writing—original draft, review, and editing.
T Uebanso: validation, investigation, and writing—original draft, review, and editing.
W Nishimura: validation, investigation, and writing—original draft, review, and editing.
M Hiramoto: validation, investigation, and writing—original draft, review, and editing.
I Shimomura: resources, validation, investigation, and writing—original draft, review, and editing.
K Yasuda: conceptualization, resources, data curation, software, formal analysis, funding acquisition, supervision, validation, investigation, methodology, project administration, and writing—original draft, review, and editing.

### Conflict of Interest Statement

The authors declare that they have no conflict of interest.

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
