## [Reviewer comments · Life Science Alliance]

Single-housing induced islet epigenomic changes are related to polymorphisms in diabetic KK mice

Takao Nammo, Nobuaki Funahashi, Haruhide Udagawa, Junji Kozawa, Kenta Nakano, Yukiko Shimizu, Tadashi Okamura, Miho Kawaguchi, Takashi Uebanso, Wataru Nishimura, Masaki Hiramoto, Ichihiro Shimomura and Kazuki Yasuda

DOI: <https://doi.org/10.26508/lsa.202302099>

Corresponding author(s): Dr. Kazuki Yasuda (National Center For Global Health and Medicine)

Review Timeline:

Submission Date:	2023-04-18
Editorial Decision:	2023-06-19
Revision Received:	2024-02-29
Editorial Decision:	2024-04-12
Revision Received:	2024-05-17
Accepted:	2024-05-20

Transaction Report:

June 19, 2023

Re: Life Science Alliance manuscript #LSA-2023-02099-T

Dr. Kazuki Yasuda

Department of Metabolic Disorder, Diabetes Research Center, Research Institute, National Center for Global Health and Medicine

Department of Metabolic Disorder, Diabetes Research Center, Research Institute, National Center for Global Health and Medicine

1-21-1 Toyama

Shinjuku-ku, Tokyo 162-8655

Japan

Dear Dr. Yasuda,

Thank you for submitting your manuscript entitled "Single-housing induced islet epigenomic changes were related to polymorphisms in diabetic KK mice" to Life Science Alliance. The manuscript was assessed by expert reviewers, whose comments are appended to this letter. We invite you to submit a revised manuscript addressing the Reviewer comments.

Thank you for this interesting contribution to Life Science Alliance. We are looking forward to receiving your revised manuscript.

Sincerely,

Novella Guidi, PhD

Scientific Editor

Life Science Alliance

B. MANUSCRIPT ORGANIZATION AND FORMATTING:

Reviewer #1 (Comments to the Authors (Required)):

Nammo et al describe an original study design that compares single housed vs control diabetes-prone KK mice, and examine the impact on metabolic phenotypes and chromatin states in pancreatic islets. They report that single housing influenced glycemia, insulin secretion and active pancreatic islet regulatory elements. Interestingly, orthologous human regions were particularly enriched in genetic variants underlying fasting glucose adjusted for BMI.

The study also provides data on how strain-specific active pancreatic islet regulatory elements are strongly associated with sequence variation, whereas this is not observed in regulatory elements induced by single housing. Interestingly, they found a particularly strong enrichment of variants flanking regulatory elements. More evidence is needed to show that this enrichment is specific for these regions, rather than other well controlled genomic regions. Furthermore, the manuscript would increase its interest if the two types of analysis (SH, and KK vs C57) are tied into a more visible overarching question. An interesting outcome of this analysis is that KK specific regions, as well as SH regions were enriched for T2D variants.

In general the manuscript needs to improve in terms of clarity and interpretation, as well as from developing a more coherent structure. Several specific suggestions for methodological improvements are also needed.

Specific points:

The rationale for using the SH model needs to be further developed in the introduction and abstract. In the 3rd paragraph, for example, there are several sentences that do not follow a perfect reasoning.

Why are the insulin results in Fig 1D and 1F completely different. Even if Figure 1D is non fasting, the order of magnitude is very different, and the results appear to be contradictory at first glance.

It is quite unexpected that more than 40% of k27ac are increased in the SH group. Authors should provide some indication that this is not technical (e.g. controlling for coverage, background, efficiency/quality of Chips, etc)

" we empirically defined a median of 49 kb non-redundant long-range cis- regulatory elements (LCREs) for each gene " Presumably the "kb" is a typo. However all of these sentences are difficult to follow.

"Given the close relationship between the SH-induced H3K27ac decrease and human diabetes (Supplemental Fig. 7) " How do decreased K27ac relate to human diabetes? This cannot be left to the reader to figure this out by digging into a suppl figure.

"Sequence disruption of critical TF binding motifs was unexpectedly unprevailing in SH- induced H3K27ac decrease in KK mice". This outcome is not necessarily expected. Many genetic effects that underlie the KK diabetes prone phenotype are likely to act elsewhere, while many changes in K27ac are likely to be secondary. It is not even clear whether there is an enrichment of disruptive sites relative to expectation, although that would not necessarily be a requirement for something like this to happen.

P9 "Since the phenotype of KK-SH mice was heritable, " unclear what the authors mean by this.

Concerning the TF motifs in strain-specific elements, are the TF mRNAs differentially expressed?

"We found that 89.9% (5,222/5,806) of the H3K27ac regions contained at least one of the known motifs, including FOXM1 and LHX2"

This type of finding suggests that the motifs have low complexity, and might not be very predictive of true binding.

The authors should also spell out what these regions are.

It would be important to know if there is enrichment for variants that disrupt the predicted affinity of enriched motifs

"Next, we focused on the SH-induced differences in the H3K27ac regions" What is the specific question here? SH-induced differences in the H3K27ac regions have already been examined in earlier sections.

"Although they were complicated by various directions of change (Fig. 6A), the variant density was significantly higher in the LCREs where at least one decreased H3K27ac region was included" How many H3K27ac regions can be included in a LCRE? Which variants exactly? (KK or KK vs C57?). This analysis seems somewhat discordant with the integration of variants with SH dependent motifs. Why is this presented separately from the rest of the SH dependent analysis of variants? Perhaps this is a different type of variant analysis? This is in any case not explained

"In line with GREAT (McLean et al. 2010)" Please explain what functionalities of GREAT were used and how.

Figure 6E, figure legend. "two types of repressive epigenomic changes..." Why are these changes repressive? What are the named genes? 1 is alpha cell specific, 2 are duct specific, why are these highlighted in particular? According to the text 10 are "candidate genes for islet dysfunction", why? Next sentence also alludes to "repressive H3K27ac changes", these are not necessarily "repressive" events

Minor points:

The KK Model, described as polygenic and inbred, should be explained more clearly in the introduction because it is not a very common strain.

Fig 1b-d provide labels for the two colors

"11-week-old KK-SH mice showed symptoms" the authors surely mean metabolic features or metabolic changes

Why do the authors carry out de novo motif analysis if they will later discard them if they do not match expected consensus motifs?

"by disruption of the transcription factor (TF)-binding motif" should be plural "of TF binding motifs"
Table 1 could be reduced to 10-5 without losing useful information

Reviewer #2 (Comments to the Authors (Required)):

Environmental factors are known to influence the epigenomic landscape, but more detailed studies are required. In this very straight forward but compelling study, the investigators addressed the influence of living alone on type 2 diabetes risk in a murine model. The findings are very notable, and I have no concerns about how this study was conducted or interpreted. The only additional limitation I would ask the authors to list is that the community is still not entirely sure what the causal effector genes are at each type 2 diabetes GWAS locus, so taking the annotated name, principally based on nearest gene, is an approximation leveraged in this current study.

Responses to the comments of Reviewer #1

- Nammo et al describe an original study design that compares single housed vs control diabetes-prone KK mice, and examine the impact on metabolic phenotypes and chromatin states in pancreatic islets. They report that single housing influenced glycemia, insulin secretion and active pancreatic islet regulatory elements. Interestingly, orthologous human regions were particularly enriched in genetic variants underlying fasting glucose adjusted for BMI.
- The study also provides data on how strain-specific active pancreatic islet regulatory elements are strongly associated with sequence variation, whereas this is not observed in regulatory elements induced by single housing. Interestingly, they found a particularly strong enrichment of variants flanking regulatory elements. More evidence is needed to show that this enrichment is specific for these regions, rather than other well controlled genomic regions. Furthermore, the manuscript would increase its interest if the two types of analysis (SH, and KK vs C57) are tied into a more visible overarching question. An interesting outcome of this analysis is that KK specific regions, as well as SH regions were enriched for T2D variants.
- In general, the manuscript needs to improve in terms of clarity and interpretation, as well as from developing a more coherent structure. Several specific suggestions for methodological improvements are also needed.

We thank the reviewer for his/her helpful comments and constructive suggestions. We have substantially improved our manuscript thanks to these valuable comments.

- **Specific points:**
- **The rationale for using the SH model needs to be further developed in the introduction and abstract. In the 3rd paragraph, for example, there are several sentences that do not follow a perfect reasoning.**

We thank the reviewer for pointing this out. According to this reviewer's comment, we first corrected the first sentence in the abstract as follows:

Page 3, lines 1-2: **Poorer social relationships are increasingly recognized as matters affecting type 2 diabetes (T2D).**

Further, we have added some description to the sentences in the 3rd paragraph as follows (characters in red):

Page 5, lines 2-13: **Furthermore**, environmental factors, including diet, have been reported to play a role in the development of diabetes. **Notably, as one of the environmental factors, the impact of social isolation on early mortality has recently been highlighted (Holt-Lunstad et al. 2010). Among various measurements for the social isolation, living alone possesses objective characteristics and has also been reported to be a risk factor for T2D in humans (Meisinger et al. 2009; Hilding et al. 2015). Interestingly, a similar situation has been found in KK mice, where single-housing is the alternative environmental manipulation accelerating the onset and development of diabetes even when fed a regular chow diet (Matsuo et al. 1971). In addition to this, KK mice are less likely to develop diabetes if they are not exposed to risk factors (Matsuo et al. 1971), as was the case with humans living in the early 1960s (Ogurtsova et al. 2017). Thus, a**

thorough understanding of the molecular mechanisms of this mouse model could be a key to addressing the global health problem arising from isolation.

- **Why are the insulin results in Fig 1D and 1F completely different. Even if Figure 1D is non fasting, the order of magnitude is very different, and the results appear to be contradictory at first glance.**

We understand the reviewer's concern about the big difference in the effect of single housing on plasma insulin levels between fasting and non-fasting KK mice. However, a recent study conducted by Oduori et al. (*J Clin Invest* 130:6639-6655,2020, PMID: 33196462, DOI: 10.1172/JCI140046) comes into our mind, because it can give a clue to understand our data. Although persistent membrane depolarization was one of the characteristic features of pancreatic beta cells derived from the beta-cell-specific *Kcnj11*-KO mice ($\beta Kcnj11^{-/-}$ mice), the authors discovered that the genetic background of KK mice can cause persistent depolarization of pancreatic beta cells in obese KK-Ay mice. Our independent evaluation also indicated evidence of persistent depolarization using isolated islets from 11-week-old KK-SH mice by the insulin secretion at levels comparable between low (2.8 mM) and high (16.7mM) glucose by static incubation (Supplemental Figure For Reviewer Only 1), similarly to the findings in $\beta Kcnj11^{-/-}$ and KK-Ay islets (as presented in the Figure 3C and 5F in the aforementioned *J Clin Invest* article, respectively). In Figure 1B in that article, the authors clearly indicated that fasting plasma insulin levels in the $\beta Kcnj11^{-/-}$ mice of persistent depolarization model were comparable to those in the control $\beta Kcnj11^{fl/fl}$ mice, supporting our data of fasting plasma insulin levels in Fig. 1F. Regarding our data of high levels of non-fasting plasma insulin in Fig. 1D, we consider that persistent depolarization can enhance compensatory increase in plasma insulin levels for insulin resistance due to larger weight gain, since blood glucose levels in KK-SH mice were significantly higher than those in KK-GH mice at the age of 11 weeks as indicated in Fig. 1C. Thus, our data are seemingly contradictory, but actually are related to each other.

- **It is quite unexpected that more than 40% of k27ac are increased in the SH group. Authors should provide some indication that this is not technical (e.g. controlling for coverage, background, efficiency/quality of Chips, etc).**

We understand the need to present the results of quality control (QC) steps of our next generation sequencing data of H3K27ac ChIP-Seq. For comparison, we also performed the same analysis for the publicly available raw sequencing data that were recently generated by an independent lab (*J Clin Invest.* 130:6639-6655, 2020, PMID: 36821378, DOI: 10.1172/JCI165208).

Please find the documents (Supplementary Figure For Reviewer Only 2) attached for reference. Our data are as follows: #1. ERR2538131 (C57BL/6J, control for high-fat diet fed mice), #2. KK-GH (Group-housed) and #3. KK-SH (Singly housed). The data collected are as follows: #4. SRR9840900 (Fed (4h) Replicate 1), #5. SRR9840901 (Fed (4h) Replicate 2), #6. SRR9840920 (Control for Week3 Lsd1 KO Replicate 1), #7. SRR9840921 (Control for Week3 Lsd1 KO Replicate 2), #8. SRR9840943 (db/+ age-matched control for db/db Replicate 1), #9. SRR9840944 (db/+ age-matched control for db/db Replicate 2), #10. SRR20673414 (Control for Week3 Lsd1 KI Replicate 1) and #11. SRR20673415 (Control for Week3 Lsd1 KI Replicate 2).

First, to find fundamental problems in high-throughput sequencing datasets, we used FastQC (Available online at: <http://www.bioinformatics.babraham.ac.uk/projects/fastqc>). Among all the 11 items checked, potentially problematic signs, including Warn and Fail, could be identified for 3 items, including per sequence GC content, sequence duplication levels and overrepresented sequences (Supplementary Figure For Reviewer Only 2 and 3). Overall, the QC results show that our datasets have a better quality with minimum problems.

Next, we also tested the quality of the aligned data (BAM files) from our ChIP-seq experiments using ChIPQC (Front Genet 5:75, 2014, PMID: 24782889, DOI: 10.3389/fgene.2014.00075) on the R platform. For the pipeline of processing raw sequencing fastq files, the data was initially filtered for blacklisted regions for mm10 (<https://github.com/Boyle-Lab/Blacklist/blob/master/lists/mm10-blacklist.v2.bed.gz>) and the H3K27ac peaks were called using MACS3 (Genome Biol 9:R137, 2008, PMID: 18798982, DOI: 10.1186/gb-2008-9-9-r137) with options "callpeak -f BAM -g mm -B -q 0.05." Following this, we could validate our H3K27ac ChIP-Seq experiments by comparison with the reported data. Among samples tested, ours showed that the proportion of genomic regions at greater depth was high enough (Supplemental Figure For Reviewer Only 4). Furthermore, the enrichment profile of reads across known and specific genomic features was very similar among samples tested (Supplemental Figure For Reviewer Only 5).

Together with these data, we found that our H3K27ac ChIP-Seq experiments were successful, and the result that more than 40% of H3K27ac regions are increased in the SH group was supported.

- **" we empirically defined a median of 49 kb non-redundant long-range cis-regulatory elements (LCREs) for each gene " Presumably the "kb" is a typo. However, all of these sentences are difficult to follow.**

We thank the reviewer for pointing this out. We re-examined the genomic intervals of LCREs from the beginning and re-confirmed that the median size was 49kb. GREAT originally proposed three kinds of LCREs, among which they used "basal plus extension rule" for the default settings. By comparison, we also found that the median size of LCREs was 197kb by this default setting. Although the difference in size distributions was very significant between the two types of LCREs with the upper limit of approximately 2Mb from the definition for both types of LCREs, the results of functional annotation tests were known to be similar to each other as reported previously (Nat Biotechnol 28:495-501, 2010, PMID: 20436461, DOI: 10.1038/nbt.1630). We are afraid that our data could be confusing because of the log-transformed y-axis (Fig. 2C). To facilitate better understanding, we prepared a new violin plot of LCREs having a vertical axis representing their genomic intervals with raw values (Supplemental Figure For Reviewer Only 6).

- **"Given the close relationship between the SH-induced H3K27ac decrease and human diabetes (Supplemental Fig. 7) " How do decreased K27ac relate to human diabetes? This cannot be left to the reader to figure this out by digging into a suppl figure.**

We apologize for the inconvenience in our original manuscript. We moved the information in Supplemental Fig. 7 to new Figure 3. We hope that the revised figures would facilitate understanding these data.

- **"Sequence disruption of critical TF binding motifs was unexpectedly unprevailing in SH- induced H3K27ac decrease in KK mice". This outcome is not necessarily expected. Many genetic effects that underlie the KK diabetes prone phenotype are likely to act elsewhere, while many changes in K27ac are likely to be secondary. It is not even clear whether there is an enrichment of disruptive sites relative to expectation, although that would not necessarily be a requirement for something like this to happen.**

We would like to thank the reviewer for the profound insights. We understand that the outcome that we observed was not necessarily expected and thus we deleted the word "unexpectedly" from the sentence (page 8, para 3).

Within the regulatory elements showing SH-induced H3K27ac decrease, we further tested the distribution of small variants in the critical TF motifs. We used the HOMER software to identify DNA sequences of known TF motifs in the reference C57BL/6J genome having a score above the pre-optimized detection threshold and found 17,438 FOXA2, 21,654 FOX-Ebox, 1,236 HNF1A, 3,284 HNF1B, 11,482 NEUROD1 and 15,219 RFX6 motifs. There were 1.05-3.62% of the TF motifs containing small variants and statistical analysis indicated that both SNVs and indels of KK mice were significantly depleted in each of the motifs relative to the genome-wide distribution, providing another insight that the base pair substitutions caused by small variants are unlikely to play a primary role for most decreases in H3K27ac level. However, we decided to further examine the small variants identified within TF motifs, according to the reviewer's comment.

As suggested by the reviewer, we predicted whether a variant was disruptive to a TF motif by assessing the biophysical models for the interaction between TFs and DNA. To this end, we constructed models on publicly available HT-SELEX datasets using a software named No Read Left Behind (NRLB; Proc Natl Acad Sci U S A. 115:E3692-E3701, 2018, PMID: 29610332, DOI: 10.1073/pnas.1714376115) (Supplemental Figure For Reviewer Only 7 and 9). We had to limit our analysis to SNVs, because indels are rarer than SNVs and could cause unavoidable complexity arising from the wide range of base compositions.

Among known TF motifs that were identified, there was a variety of binding affinity to the TF of interest. We observed up to three base pair substitutions in these motifs in KK mice, although substitutions of two or more base pairs were found to be extremely rare. The estimation from all possible variations of identified TF motifs revealed that the less base pair substitutions a TF motif accumulated, the more motifs were expected to show increases in the binding affinity. Despite this, single base pair substitutions were expected to be disruptive for more than 75% of the motif alterations (Supplemental Figure For Reviewer Only 8B and C). In the case of observed variation in KK mice, although single base pair substitutions were found to cause affinity disruption for similar proportions for all kinds of motifs tested, the enrichment of disruptive sites relative to expectation was statistically significant only for NEUROD1 motifs (Supplemental Figure For Reviewer Only 8B). These results suggest that natural base pair substitutions in the TF motifs are overall disruptive and they are under natural selection with various directions depending on the kind of TF motifs. From these results, we added a sentence in the text as follows:

Page 9, lines 12-14: **We found that more than 75% of such overlaps were disruptive to the binding affinity as estimated by the "No Read Left Behind" energy prediction algorithm (Rastogi et al. 2018) (Supplementary Figure For Reviewer Only 8B) and could indeed be involved in the SH-induced local decrease in H3K27ac.**

Although we agree with the reviewer's idea that enrichment of disruptive sites would not be the requirement for SH-induced H3K27ac changes, it would be a subject of future study to investigate the mechanisms of the variants flanking regulatory elements.

- **P9 "Since the phenotype of KK-SH mice was heritable, " unclear what the authors mean by this.**

Regarding this point, we have added more description in the text as follows:

Page 9, lines 1-5: Since the SH-induced weight gain and diabetes was a set of phenotypes that was transmissible from parent to child in KK mice, it could be possible to identify the associated changes in the genomic DNA sequences relative to the reference genome of C57BL/6J mice. We first tested whether sequence variants could be involved in the H3K27ac decrease caused by nucleotide change(s) in critical TF motifs (Fig. 3B; Supplemental Fig. 7).

- **Concerning the TF motifs in strain-specific elements, are the TF mRNAs differentially expressed?**

We agree with this reviewer's comment and it is possible that the differentially expressed transcription factors could be involved in the presence or absence of the strain-specific H3K27ac regions. To be precise, it would be necessary to compare the expression levels of Lhx2 and Foxm1 mRNAs when strain-specific H3K27ac enrichment started in specific tissues or cell types during development, however, such data are currently not available. For reference purposes only, we tentatively performed a comparative analysis of our whole transcriptome data between chow-fed 54-week-old C57BL/6J and 11-week-old KK-GH mice. We found that, in pancreatic islets, Lhx2 was not expressed and that Foxm1 was not differentially expressed (Supplemental Figure For Reviewer Only 9). However, this does not deny the possibility that differential expression of TF mRNAs caused the strain-specific elements, which we consider would be a research question to be investigated in the future.

- **"We found that 89.9% (5,222/5,806) of the H3K27ac regions contained at least one of the known motifs, including FOXM1 and LHX2"**
- **This type of finding suggests that the motifs have low complexity, and might not be very predictive of true binding.**
- **The authors should also spell out what these regions are.**
- **It would be important to know if there is enrichment for variants that disrupt the predicted affinity of enriched motifs.**

We thank the reviewer for the careful review of the manuscript. Since our study focused on TF motifs identified using a single PWM model and did not consider true TF binding events, we have discussed this issue as one of the limitations of this study in the DISCUSSION section (page 14, para 3) as follows.

Page 14, lines 24-26: **Third, the TF motifs identified using a single position weight matrix model have low complexity, and might not be very predictive of true binding.**

With regard to the C57BL/6J strain-specific regulatory elements, we presented the data of GREAT functional enrichment analysis in Supplemental Table S13. Although this included some enriched terms associated with endocrine pancreas function and glucose utilization, the overall results still allow a relatively broad interpretation. To obtain a more specific view, we performed

the GREAT analysis again by changing the filter of binomial fold enrichment (set to ≥ 2 -fold by default). With a more moderate threshold of ≥ 1.5 -fold, we found that the terms related to insulin secretion and action predominantly dominated the top 10 rankings, which was followed by pathological terms related to diabetes and obesity, including abnormal fatty acid level, abnormal hepatocyte morphology and abnormal fetal growth/weight/body size. Based on these observations and Supplemental Fig XX, we have come to understand that while the set of C57BL/6J strain-specific regulatory elements is not a complete collection of susceptibility loci for diabetes and the related diseases on their own, they have the potential to form a fully connected cis-regulatory network when joined by other complementary elements.

With regard to the enrichment for the disruptive variants, we performed the same analysis for the distribution of small variants within the identified TF motifs as described earlier for the reviewer's comment (Supplemental Figure For Reviewer Only 10 and 11). Using the HOMER software, we found 12,337 FOXM1 and 10,906 LHX2 motifs in the C57BL/6J strain specific H3K27ac regions, among which we found that 1.49-2.48% of the TF motifs harbored small variants. We found a significant depletion of SNVs both in FOXM1 and in LHX2 motifs, although indels were significantly enriched in the FOXM1 motifs and were significantly depleted in the LHX2 motifs, thus suggesting the enrichment of disruptive indels in the FOXM1 motifs.

We also predicted whether an SNV was disruptive to a TF motif using NRLB (No Read Left Behind; Proc Natl Acad Sci U S A. 115:E3692-E3701, 2018, PMID: 29610332, DOI: 10.1073/pnas.1714376115). As observed earlier, the estimation from all possible variations of identified known FOXM1 or LHX2 motifs revealed that the less base pair substitutions a TF motif accumulated, the more motifs were expected to show increases in the binding affinity (Supplemental Figure For Reviewer Only 11C). We also observed that single base pair substitutions were expected to be disruptive for more than 75% of the motif alterations. In the case of observed variations in KK mice, although single base pair substitutions were found to cause affinity disruption for similar proportions both for FOXM1 motifs and for LHX2 motifs, both motifs revealed the significant underrepresentation of disruptive sites relative to expectation (Supplemental Figure For Reviewer Only 11B). Therefore, these data suggest that natural base pair substitutions in the TF motifs of FOXM1 or LHX2 are overall disruptive and they are under natural selection with various directions. According to these results, we added sentences in the text as follows:

Page 10, lines 21-24: *As described above for the SH-induced H3K27ac decrease, we estimated using NRLB that more than 75% of such overlaps were found to be disruptive to the motif affinity, and thus could cause the pre-determined local H3K27ac absence. However, in most cases of KK-selective absence in H3K27ac, sequence disruption of enriched motifs is also unlikely to be a direct cause.*

- **"Next, we focused on the SH-induced differences in the H3K27ac regions" What is the specific question here? SH-induced differences in the H3K27ac regions have already been examined in earlier sections.**

We thank the reviewer for pointing out our ambiguous description. The specific question here is whether there are roles of sequence variants outside H3K27ac in the H3K27ac response to SH. Regarding this, we have revised our sentences in the text as follows:

Page 10, line 25-Page 11, line 1: Since the density of sequence variants was significantly higher outside than inside H3K27ac (Fig. 4F; Supplemental Fig. 8C), we subsequently asked whether

sequence variants outside H3K27ac had any impact on the various types of H3K27ac changes in pancreatic islets of KK mice.

Page 11, lines 8-10: Next, given the relatively little role of sequence variants inside H3K27ac in the SH-induced H3K27ac changes as already mentioned, we investigated whether sequence variants outside H3K27ac had any impact on the H3K27ac response to SH.

- "Although they were complicated by various directions of change (Fig. 6A), the variant density was significantly higher in the LCREs where at least one decreased H3K27ac region was included" How many H3K27ac regions can be included in a LCRE?
- Which variants exactly? (KK or KK vs C57?). This analysis seems somewhat discordant with the integration of variants with SH dependent motifs. Why is this presented separately from the rest of the SH dependent analysis of variants? Perhaps this is a different type of variant analysis? This is in any case not explained.

Our analysis revealed that up to 45 H3K27ac regions were included in a 556-kb-long LCRE that was assigned to the *Ptprn2* gene, whereas up to 75 H3K27ac regions were identified in the same LCRE using the same peak calling method for the downloaded H3K27ac ChIP-Seq data. Given that the FastQC results of our H3K27ac ChIP-Seq reads showed significantly lower %GC and less deviations from the theoretical normal distribution than the other downloaded data (Supplemental Figure For Reviewer Only 3), we speculate that the lower maximum numbers of H3K27ac regions in our samples are due to the successful detection of short read coverages in the regulatory regions with lower GC content, resulting in the moderately elongated H3K27ac regions without being fragmented into smaller pieces. We believe that this idea was supported by the observation that the average GC content of whole H3K27ac regions in the *Ptprn2*-LCRE was significantly lower in our samples than that in the others (45.7% vs 46.6%; $p < 0.05$, two-tailed Welch's *t*-test), thus validating our experiments.

With regard to the term "variants" used in that sentence, we used it in the sense of genetic differences in KK mice relative to the mouse reference genome of C57BL/6J, which was the initial sequence firstly reported in 2002 (Nature 420:520-562, 2002, PMID: 12466850, DOI: 10.1038/nature01262) to which all subsequent sequences for other mouse strains were compared. To facilitate understanding, we made a correction to that sentence as follows:

Page 11, lines 10-13: Although they were complicated by various directions of change (Fig. 6A), the variant density of KK mice in the setting of mm10 assembly was significantly higher in the LCREs where at least one decreased H3K27ac region was included (Fig. 6B; Supplemental Fig. 9C; Supplemental Tables 6 and 7).

With regard to the analysis of variant density in LCREs, we found it useful to incorporate the reviewer's suggestion about the controlled genomic regions raised at the beginning of his/her comments. To perform the genomic segmentation using the unsupervised learning method named ChromHMM (Nat Protoc 12:2478-2492, 2017, PMID: 29120462, DOI: 10.1038/nprot.2017.124), we defined an 18-chromatin-states model by integrating ChIP-Seq data of the 6 most fundamental kinds of histone modification as controls, including H3K4me1, H3K4me3, H3K27me3, H3K9me3, H3K36me3 and H3K27ac (Supplemental Figure For Reviewer Only 12). By this, we found that the small variants in KK mice were rather enriched in the genomic regions that lack these modifications, supporting our data indicating a strong enrichment of variants flanking regulatory elements. Because we thought it was inadequate to discuss such small variants in relation to the TF binding event in the accessible genomic regions, we made a separate presentation of the data to develop a novel concept of variant function.

- **"In line with GREAT (McLean et al. 2010)" Please explain what functionalities of GREAT were used and how.**

We thank the reviewer for pointing out our incomplete explanation about the data analysis. The concept proposed by GREAT (Nat Biotechnol 28:495-501, 2010, PMID: 20436461, DOI: 10.1038/nbt.1630) was epoch-making in that it could include distal cis-regulatory elements for the functional enrichment analysis by making use of a binomial test, whereas the conventional methods could use only proximal elements by using a hypergeometric test. Since using only binomial test can cause false-positive results arising from a gene-specific enrichment of proximal and distal elements rather than from the enrichment involving multiple genes associated with a certain term, the GREAT output by default comprises only enriched terms significant by both binomial and hypergeometric tests. In our analysis, however, we decided to pick up all the results that were significant only for the binomial test. According to the above literature, we also selected the "single nearest gene" rule from three options for the definition of LCREs and the susceptible genomic loci for various GWAS diseases/traits, so that all the distal information about H3K27ac regions could be included in the enrichment analysis. Hypergeometric test was not included, since we were interested in covering both types of results with term-originated and gene-specific enrichment. Furthermore, we filtered the GWAS diseases/traits with number of associated genes still less than 5 in order to focus on more robust evidence. According to this reviewer's comment, we made some corrections to the text as follows:

Page 12, lines 3-7: **Having been motivated by the analytical strategy of GREAT (McLean et al. 2010), we utilized the advantages of a binomial test to assess the enrichment of susceptible diseases/traits in the H3K27ac regions in both proximal and distal elements, by which the output results included both term-derived and gene-specific enrichment (see methods).**

Page 19, lines 21-27: **We took advantage of the concept of GREAT (McLean et al. 2010) in terms of gene assignment and used only a binomial test, although the GREAT output by default comprises only enriched terms significant by both binomial and hypergeometric tests to avoid false-positive results arising from a gene-specific enrichment of proximal and distal H3K27ac elements rather than from the enrichment involving multiple genes associated with a certain term. We picked up all the results that were significant only for the binomial test, since we were interested in covering both types of results with term-originated and gene-specific enrichment.**

Although we were able to obtain some reasonable results by our analytical approach of assuming candidate effector genes in each GWAS loci, it should be noted that another reviewer critically expressed a concern about the uncertainty associated with this speculation. According to his/her constructive suggestion, we have added the below statement as one of the limitations of this study.

Page 14, line 27-Page 15, line 3: **Fifth, since the causal effector genes at each type 2 diabetes GWAS locus have not been determined yet, we have to say that our approach of taking the disease annotation, principally based on nearest genes, is just an approximation.**

- **Figure 6E, figure legend. "two types of repressive epigenomic changes..." Why are these changes repressive? What are the named genes? 1 is alpha cell specific, 2 are duct specific, why are these highlighted in particular? According to the text 10 are**

"candidate genes for islet dysfunction", why? Next sentence also alludes to "repressive H3K27ac changes", these are not necessarily "repressive" events.

We agree with the reviewer's comment that the two types of epigenomic changes, including KK-selective absence and SH-induced decrease, are not necessarily repressive, and we have made corrections to the quoted sentence as follows:

Page 34, lines 9-11: Two types of **insufficient H3K27ac status in the pancreatic islets of KK mice** converge to expand the number of downregulated type 2 diabetes susceptibility genes under single-housing condition.

With regard to the differentially expressed genes listed in the Fig. 6E, we initially intended to select candidate genes for islet dysfunction and highlighted some of them with the aid of information of both cell-type specificity and expression levels from the single-cell RNA-Seq datasets (<https://sandberglab.se/tool/pancreas>, Cell Metab 24:593-607, 2016, PMID: 27667667, DOI: 10.1016/j.cmet.2016.08.020). However, according to the reviewer's constructive comment, we realize that such information will not provide a direct explanation to the islet dysfunction and pruning the gene list could cause a loss of potential data. Therefore, we simply listed all the 16 downregulated genes harboring either or both of two types of insufficient H3K27ac status in KK-SH mice, and suggested relevant published articles for reference as follows.

We further made corrections to the sentences in the text where we inappropriately used the word "repressive" as follows:

Page 12, lines 20-26: Although the highest enrichment was observed for both SH-induced H3K27ac decrease and KK-selective H3K27ac absence, a large proportion (~30%) of associations with T2D loci were provided by either of the two types of **insufficient H3K27ac levels** (Fig. 6E), suggesting an environmental role in the T2D pandemic (Ogurtsova et al. 2017; Zheng et al. 2018). Of the 467 T2D GWAS genes harboring **such insufficient H3K27ac**, **16** genes were found to be significantly downregulated in **pancreatic islets of KK-SH mice**, which showed different patterns of association with either or both types of **insufficient H3K27ac levels** (Fig. 6E; Supplemental Table 1).

- **Minor points:**
- **The KK Model, described as polygenic and inbred, should be explained more clearly in the introduction because it is not a very common strain.**

According to the reviewer's constructive comment, we added some descriptions in the introduction as follows:

Page 4, lines 18-25: **KK mice are an inbred strain originally derived from experimental albino mice obtained in Kasukabe district in the suburbs of Tokyo, Japan (Kondo 1957; Ikeda 1994). Initially, it had been established through a selection regime for a reduced variance in body weight, gentle nature and high reproductivity to produce the ideal experimental mouse line (Kondo 1957), but was reported to be hereditarily obese and diabetic several years later (Nakamura 1962). Although it is regarded as a model of diabetes because of insulin resistance due to mild obesity (Dulin and Wyse 1970), defects in the insulin secretory response to glucose and pathological changes in beta cells exist (Ikeda 1994).**

Page 4, line 26-Page 5, line 3: **By the beginning of the 2000's, genetic approaches, such as QTL mapping, identified several susceptibility loci associated with diabetes and related diseases in KK mouse, suggesting** its polygenic nature (Suto et al. 1998; Shike et al. 2001). **Furthermore,** environmental factors, including diet, have been reported to play a role in the development of diabetes (Matsuo et al. 1971).

- **Fig 1b-d provide labels for the two colors.**

In Fig. 1B-D, we showed time-course analysis of body weight, non-fasting blood glucose levels and non-fasting plasma insulin levels. The data were shown using either bars or boxes in each graph for comparison of two categories, group-housed and singly housed KK mice, which we colored green and red, respectively. To facilitate distinguishing between the labels, we have changed the figure legend as follows.

Page 31, lines 5-14: (B) Body weight **of KK-GH (green bars) and KK-SH (red bars) mice.** KK-GH, n=11 (9- to 11-week-old) or 10 (13- to 15-week-old); KK-SH, n=11 (9- to 11-week-old) or 10 (13- to 15-week-old). (C) Non-fasting glycemia **of KK-GH (green bars) and KK-SH (red bars) mice.** KK-GH, n=11 (9-week-old) or 10 (11- to 15-week-old); KK-SH, n=11 (9-week-old) or 10 (11- to 15-week-old). The data for mice at 9 weeks old were obtained 2 days after grouping. n.d., not done. A linear trend test was performed to determine age-dependent increase in blood glucose levels for each group. p values were calculated using multiple t tests using the two-stage step-up false discovery rate method of Benjamini, Krieger, and Yekutieli. (D) Non-fasting insulin **of KK-GH (green boxes) and KK-SH (red boxes) mice.** KK-GH, n=10; KK-SH, n=10. The data for mice at 9 weeks old were obtained 2 days after grouping.

- **"11-week-old KK-SH mice showed symptoms" the authors surely mean metabolic features or metabolic changes.**

We thank the reviewer for the careful review of the manuscript. We rewrote the quoted phrase to "11-week-old KK-SH mice showed **metabolic features**" (page 6, para 1).

- **Why do the authors carry out de novo motif analysis if they will later discard them if they do not match expected consensus motifs?**

Although de novo motif analysis is great in that it provides unbiased identification of the most over-represented motifs from a large number of divergent DNA sequences, it has also been pointed out that the strategy of estimating known motif enrichment can outperform it for such datasets showing less degree of freedom in the DNA sequences, as illustrated by the examples of PU.1 (GAGGAAGT) and ISRE (GAAACTGAAA) motifs harbored in the GA-rich sequences (<http://homer.ucsd.edu/homer/introduction/basics.html>). Our dataset of SH-induced H3K27ac increases revealed preferential localization surrounding TSSs (Supplemental Fig 4A), which we guess to be the reason for the de novo motifs with less similarity to the canonical sequences due to the GC-rich background. Therefore, we performed the known motif analysis and found a strong enrichment of NRF1 (CTGCGCATGCGC) motif in those elements. Regarding this point, we added some explanation in the text as follows:

Page 19, lines 6-11: De novo motif enrichment for the differential H3K27ac regions was also assessed relative to the custom background sequences of KK mice using HOMER (Heinz et al. 2010) with the command "findMotifsGenome.pl" with options -size given -mask -mset

vertebrates, which was also useful for the identification of the enrichment of known canonical TF motifs especially in the DNA sequences with less degree of freedom (<http://homer.ucsd.edu/homer/introduction/basics.html>).

- **"by disruption of the transcription factor (TF)-binding motif" should be plural "of TF binding motifs**

Table 1 could be reduced to 10-5 without losing useful information

We thank the reviewer for pointing these matters. We changed the wording to “by disruption of the transcription factor (TF)-binding motifs” (page 4, para 1). We also corrected the Table 1 and 2, in each of which the top 5 enriched results were presented.

Responses to the comments of Reviewer #2

- **Environmental factors are known to influence the epigenomic landscape, but more detailed studies are required. In this very straight forward but compelling study, the investigators addressed the influence of living alone on type 2 diabetes risk in a murine model. The findings are very notable, and I have no concerns about how this study was conducted or interpreted. The only additional limitation I would ask the authors to list is that the community is still not entirely sure what the causal effector genes are at each type 2 diabetes GWAS locus, so taking the annotated name, principally based on nearest gene, is an approximation leveraged in this current study.**

We thank the referee for his/her helpful comments and constructive suggestions. We have substantially improved our manuscript thanks to these valuable comments.

We totally agree with the reviewer's comment that our approach of assuming effector genes based on nearest genes is just an approximation, and we have added a sentence in the text as follows:

Page 14, line 27-Page 15, line 3: **Fifth, since the causal effector genes at each type 2 diabetes GWAS locus have not been determined yet, we have to say that our approach of taking the disease annotation, principally based on nearest genes, is just an approximation.**

April 12, 2024

RE: Life Science Alliance Manuscript #LSA-2023-02099-TR

Dr. Kazuki Yasuda
National Center For Global Health and Medicine
Department of Metabolic Disorder, Diabetes Research Center, Research Institute,
1-21-1 Toyama
Shinjuku-ku, Tokyo 162-8655
Japan

Dear Dr. Yasuda,

Thank you for submitting your revised manuscript entitled "Single-housing induced islet epigenomic changes were related to polymorphisms in diabetic KK mice". We would be happy to publish your paper in Life Science Alliance pending final revisions necessary to meet our formatting guidelines.

- please pay careful attention to address Reviewer 1's comments
- please be sure that the authorship listing and order is correct
- please upload all figure files as individual ones, including the supplementary figure files; all figure legends should only appear in the main manuscript file
- please add ORCID ID for the corresponding author -- you should have received instructions on how to do so
- please add the Twitter handle of your host institute/organization as well as your own or/and one of the authors in our system
- please insert legends for all supplementary tables
- please upload all your Tables in editable .doc or excel format
- please add callouts for Figures S1A,B; S2A,B; S6B,C to your main manuscript text
- please update the Data Availability statement to remove mention of the referees

A. FINAL FILES:

B. MANUSCRIPT ORGANIZATION AND FORMATTING:

Sincerely,

Reviewer #1 (Comments to the Authors (Required)):

The authors have improved the manuscript, although despite the long explanation most of my suggestions have not been addressed. Given that other reviewers are supportive, at this point I think the main thing is that the authors and editor may wish to improve the clarity and conceptual coherence of the article.

I will not go through all points again, but will mention some examples:

- In the response regarding measures to account for differences in coverage etc to enable proper quantification of Chip-seq, the authors do not seem to have taken these type of precautions
- 49 kb presumably refers to a cis regulatory element domain, not a regulatory element. Regulatory elements are not 49 kb.
- The manuscript needs editorial improvements. The new first sentence in the abstract is a case in point " Poorer social relationships are increasingly recognized as matters affecting type 2 diabetes (T2D)". This sentence can be improved from the standpoint of grammar, style and clarity (what do the authors mean by "matters affecting T2D": are they thought to cause diabetes? Are the authors talking about how poor social relationships affect the management of diabetes?

Likewise this new sentence is hard to follow: "Two types of insufficient H3K27ac status in the pancreatic islets of KK mice converge to expand the number of downregulated type 2 diabetes susceptibility genes under single-housing condition.

This needs to be done throughout the manuscript.

Previous review item

"Given the close relationship between the SH-induced H3K27ac decrease and human diabetes (Supplemental Fig. 7) " How do decreased K27ac relate to human diabetes? This cannot be left to the reader to figure this out by digging into a suppl figure".

Response in rebuttal:

We apologize for the inconvenience in our original manuscript. We moved the information in Supplemental Fig. 7 to new Figure 3. We hope that the revised figures would facilitate understanding these data.

The reader will still not know from the text why the authors think that there is "close relationship between the SH-induced H3K27ac decrease and human"

Again, this is just an example to illustrate that the responses are insufficient

Responses to the comments of Reviewer #1

- **The authors have improved the manuscript, although despite the long explanation most of my suggestions have not been addressed. Given that other reviewers are supportive, at this point I think the main thing is that the authors and editor may wish to improve the clarity and conceptual coherence of the article.**

I will not go through all points again, but will mention some examples:

We sincerely thank the reviewer for providing helpful comments and constructive suggestions despite the incomplete responses in the last submission. We have substantially improved our manuscript based on these valuable comments. First, we would like to begin by responding to the latest comments made by Reviewer #1, followed by revised responses to the previous comments.

- **- In the response regarding measures to account for differences in coverage etc to enable proper quantification of Chip-seq, the authors do not seem to have taken these type of precautions**

We understand the importance of quality control (QC) of the H3K27ac ChIP-Seq data and initially attached importance only to the sequence duplication level; however, this is not sufficient on its own. Because the pancreatic islet is a microorgan, it was possible to collect only a small amount of high-quality ChIP samples from mice within a relatively short period of time, but we could not divide them for the preparation of ChIP-qPCR experiments; thus, we did not assess the efficiency or quality of ChIP for the local enrichment of selected elements. To make up for this lack of information, we added data of "FRiP (Fraction of reads in peaks)" from ChIPQC results (Supplemental Figure for Reviewer Only 4), which is considered to indicate a "signal-to-noise" measure and our data are > 12% (around 5% or higher is generally judged as successful enrichment). In this study, ChIP was performed using the same protocol for each sample. Thus, we consider that the data are comparable, and that it is possible to carry out a reliable differential analysis of ChIP-Seq using SICER2, which includes a normalization step before comparing the experimental and control data.

- **- 49 kb presumably refers to a cis regulatory element domain, not a regulatory element. Regulatory elements are not 49 kb.**

We thank the reviewer for this helpful comment. Certainly, the median size of the cis-regulatory elements was much smaller than 49kb, as indicated in the newly prepared Fig. 2C. Therefore, we referred to what we previously called LCRE as LCRED (long-range cis-regulatory element domain) in the latest version of the manuscript.

- **- The manuscript needs editorial improvements. The new first sentence in the abstract is a case in point**
" Poorer social relationships are increasingly recognized as matters affecting type 2 diabetes (T2D)". This sentence can be improved from the standpoint of grammar, style and clarity (what do the authors mean by "matters affecting T2D": are they thought to cause diabetes? Are the authors talking about how poor social relationships affect the management of diabetes?)

We thank the reviewer for this valuable comment. As suggested by the reviewer, we think that there may be various possibilities regarding how social relationships are related to T2D, ranging from biological mechanisms to clinical management. However, we also think that the contribution of biological mechanisms would be important given the data from our mouse study. According to the reviewer's suggestion, we rewrote the sentence as follows:

Page 3, line 49: A lack of social relationships is increasingly recognized as a type 2 diabetes (T2D) risk.

- **Likewise this new sentence is hard to follow: "Two types of insufficient H3K27ac status in the pancreatic islets of KK mice converge to expand the number of downregulated type 2 diabetes susceptibility genes under single-housing condition."**

This needs to be done throughout the manuscript.

We agree with the reviewer's comments. We have rewritten this sentence in figure legend of Fig 7E as follows. Besides, we re-counted the number of type 2 diabetes susceptibility genes associated with KK-selective absence and/or SH-induced decrease in H3K27ac, and it turned out that there were actually 10, not 16, as mentioned in the revised manuscript. We apologize for the error. In addition, for editorial improvement, we have had our sentences checked throughout the manuscript by a native English-speaking specialist in professional scientific writing.

Page 38, lines 953-955: Venn diagram showing the number of type 2 diabetes susceptibility genes associated with KK-selective absence and/or single-housing induced decrease in H3K27ac. Of all the 467 genes, ten that showed downregulated expression after single-housing are shown.

- **Previous review item**
"Given the close relationship between the SH-induced H3K27ac decrease and human diabetes (Supplemental Fig. 7) " How do decreased K27ac relate to human diabetes? This cannot be left to the reader to figure this out by digging into a suppl figure".
Response in rebuttal:
We apologize for the inconvenience in our original manuscript. We moved the information in Supplemental Fig. 7 to new Figure 3. We hope that the revised figures would facilitate understanding these data.

The reader will still not know from the text why the authors think that there is "close relationship between the SH-induced H3K27ac decrease and human"

Again, this is just an example to illustrate that the responses are insufficient

We thank the reviewer for this helpful suggestion. To facilitate the understanding of the kind of human diabetes the SH-induced H3K27ac decrease is related to, we added more information to the sentences in the previous section as follows. Moreover, we have reviewed our response letter to the reviewers' comments to make it as refined as possible.

Page 8, lines 173-181: Among the six detected motifs, four were highly similar to the known consensus motifs HNF1B ($p=10^{-38}$), NEUROG2 ($p=10^{-32}$), FOXA1 ($p=10^{-23}$), and RFXDC2 ($p=10^{-15}$) (Fig 3B), among which HNF1B is known to be responsible for maturity-onset diabetes

of the young (MODY) type 5 (Horikawa et al. 1997). Furthermore, each motif was similar to that of HNF1A, NEUROD1, FOXA2 (FOX:Ebox), and RFX6 (Fig 3B), which have been established as the causative or susceptible genes of human MODY3 (Yamagata et al. 1996), MODY6 (Malecki et al. 1999), T2D (Gaulton et al. 2015), and neonatal diabetes mellitus (Smith et al. 2010), respectively, suggesting the possible involvement of similar pathological mechanisms of human diabetes in KK-SH mice.

In addition, we deleted the word “close” from the quoted sentence to avoid overstatement regarding the relationship with human diabetes.

Page 8, lines 186-188: Given the ~~close~~ relationship between SH-induced decrease in H3K27ac and human diabetes (Fig 3B), we performed a functional analysis using GREAT to gain insights into the various H3K27ac regions (Table S9).

We also modified the response to the previous comments by Reviewer #1 as below, according to the reviewer's suggestion. We hope this will clarify the important points.

Responses to the comments of Reviewer #1

- **Nammo et al describe an original study design that compares single housed vs control diabetes-prone KK mice, and examine the impact on metabolic phenotypes and chromatin states in pancreatic islets. They report that single housing influenced glycemia, insulin secretion and active pancreatic islet regulatory elements. Interestingly, orthologous human regions were particularly enriched in genetic variants underlying fasting glucose adjusted for BMI.**
- **The study also provides data on how strain-specific active pancreatic islet regulatory elements are strongly associated with sequence variation, whereas this is not observed in regulatory elements induced by single housing. Interestingly, they found a particularly strong enrichment of variants flanking regulatory elements. More evidence is needed to show that this enrichment is specific for these regions, rather than other well controlled genomic regions. Furthermore, the manuscript would increase its interest if the two types of analysis (SH, and KK vs C57) are tied into a more visible overarching question. An interesting outcome of this analysis is that KK specific regions, as well as SH regions were enriched for T2D variants.**
- **In general, the manuscript needs to improve in terms of clarity and interpretation, as well as from developing a more coherent structure. Several specific suggestions for methodological improvements are also needed.**

We thank the reviewer for the helpful comments and constructive suggestions. We have substantially improved our manuscript based on these valuable comments.

- **Specific points:**
- **The rationale for using the SH model needs to be further developed in the introduction and abstract. In the 3rd paragraph, for example, there are several sentences that do not follow a perfect reasoning.**

We thank the reviewer for highlighting this. According to the reviewer's comment, we have corrected the first sentence in the Abstract as follows:

Page 3, lines 49: A lack of social relationships is increasingly recognized as a type 2 diabetes (T2D) risk.

Furthermore, we have added a description to the sentences in the 3rd paragraph as follows:

Page 5, lines 92-104: Environmental factors, including diet, have been reported to play a role in the development of diabetes (Matsuo et al. 1971). Notably, as one of the environmental factors, the impact of social isolation on early mortality has recently been highlighted (Holt-Lunstad et al. 2010). Among the various measures of social isolation, living alone possesses objective characteristics and has been reported to be a risk factor for T2D in humans (Meisinger et al. 2009; Hilding et al. 2015). Interestingly, a similar situation has been observed in KK mice, where single housing is an alternative environmental manipulation that accelerates the onset and development

of diabetes, even when mice are fed regular chow diet (Matsuo et al. 1971). In addition, KK mice are less likely to develop diabetes if they are not exposed to risk factors (Matsuo et al. 1971), as was the case for humans living in the early 1960s (Ogurtsova et al. 2017). Thus, a thorough understanding of the molecular mechanisms of this mouse model could be the key to addressing the global health problems arising from isolation.

- **Why are the insulin results in Fig 1D and 1F completely different. Even if Figure 1D is non fasting, the order of magnitude is very different, and the results appear to be contradictory at first glance.**

We understand the reviewer's concern regarding the significant difference in the effect of single housing on plasma insulin levels between fasting and non-fasting KK mice. However, a recent study conducted by Oduori et al. (*J Clin Invest* 130:6639-6655,2020; PMID: 33196462; DOI: 10.1172/JCI140046) came to our mind because it can provide clues to understanding our data. Although persistent membrane depolarization is a characteristic feature of pancreatic beta cells derived from beta cell-specific *Kcnj11*-KO mice ($\beta Kcnj11^{-/-}$ mice), the authors discovered that the genetic background of KK mice could cause persistent depolarization of pancreatic beta cells in obese KK-Ay mice. Our independent evaluation also provided evidence of persistent depolarization using islets isolated from 11-week-old KK-SH mice by insulin secretion at levels comparable between low (2.8 mM) and high (16.7mM) glucose by static incubation (Supplemental Figure for Reviewer Only 1), similar to the findings in $\beta Kcnj11^{-/-}$ and KK-Ay islets (as presented in Figure 3C and 5F in the aforementioned *J Clin Invest* article, respectively). As shown in Figure 1B in that article, the authors clearly indicated that fasting plasma insulin levels in $\beta Kcnj11^{-/-}$ mice of the persistent depolarization model were comparable to those in control $\beta Kcnj11^{fl/fl}$ mice, supporting our data on fasting plasma insulin levels in Fig 1F. Regarding our data of high levels of non-fasting plasma insulin in Fig 1D, we consider that persistent depolarization can enhance the compensatory increase in plasma insulin levels for insulin resistance due to larger weight gain, since blood glucose levels in KK-SH mice were significantly higher than those in KK-GH mice at 11 weeks of age in Fig 1C. Thus, our data are seemingly contradictory, but are actually related to each other.

- **It is quite unexpected that more than 40% of k27ac are increased in the SH group. Authors should provide some indication that this is not technical (e.g. controlling for coverage, background, efficiency/quality of Chips, etc).**

We understand the need to present the results of the quality control (QC) steps of our next-generation sequencing data from H3K27ac ChIP-Seq. For comparison, we performed the same analysis on publicly available raw sequencing data that were recently generated by an independent laboratory (*J Clin Invest.* 130:6639-6655, 2020, PMID: 36821378, DOI: 10.1172/JCI165208).

Please find the attached documents (Supplemental Table for Reviewer Only 1) for reference. Our data were as follows: 1. ERR2538131 (C57BL/6J, control for HFD-fed mice), #2. ERR2538132 (C57BL/6J, HFD-fed mice); #3. KK-GH (group-housed), #4. KK-SH (Single-housed). The following data were collected: #5. SRR9840900 (Fed (4h) Replicate 1), #6. SRR9840901 (Fed (4h) Replicate 2), #7. SRR9840920 (Control for Week3 Lsd1 KO Replicate 1), #8. SRR9840921 (Control for Week3 Lsd1 KO Replicate 2), #9. SRR9840943 (db/+ age-matched control for db/db, replicate 1), #10. SRR9840944 (db/+ age-matched control for db/db, replicate 1).

2), #11. SRR20673414 (Control for Week3 Lsd1 KI replicate 1) and #12. SRR20673415 (Control for Week3 Lsd1 KI replicate 2).

First, to identify fundamental problems in high-throughput sequencing datasets, we used FastQC (<http://www.bioinformatics.babraham.ac.uk/projects/fastqc>). Among the 11 items checked, potentially problematic signs, including Warn and Fail, could be identified for three items, including per-sequence GC content, sequence duplication levels, and overrepresented sequences (Supplemental Table for Reviewer Only 1; Supplemental Figure for Reviewer Only 2). Overall, the QC results showed that our datasets were of better quality with minimal problems.

Next, we tested the quality of the aligned data (BAM files) from our ChIP-seq experiments using ChIPQC (Front Genet 5:75, 2014; PMID: 24782889; DOI: 10.3389/fgene.2014.00075) on the R platform. For the pipeline of processing raw sequencing fastq files, the data was initially filtered for blacklisted regions for mm10 (<https://github.com/Boyle-Lab/Blacklist/blob/master/lists/mm10-blacklist.v2.bed.gz>) and the H3K27ac peaks were called using MACS3 (Genome Biol 9:R137, 2008; PMID: 18798982; DOI: 10.1186/gb-2008-9-9-r137) with options “callpeak -f BAM -g mm -B -q 0.05.” Next, we validated our H3K27ac ChIP-Seq experiments by comparing them with previously reported data. Among the samples tested, our results showed that both proportion of genomic regions at greater depths (Supplemental Figure for Reviewer Only 3) and signal-to-noise measures (Supplemental Figure for Reviewer Only 4) were sufficiently high. Furthermore, the enrichment profiles of reads across known and specific genomic features were very similar among the tested samples (Supplemental Figure for Reviewer Only 5).

Together with these data, we found that our H3K27ac ChIP-Seq experiments were successful and the result that more than 40% of H3K27ac regions were increased in the SH group was supported.

- **" we empirically defined a median of 49 kb non-redundant long-range cis-regulatory elements (LCREs) for each gene " Presumably the "kb" is a typo. However, all of these sentences are difficult to follow.**

We agree with the reviewer's comments. As suggested, we have referred to what we previously called LCRE as LCRED (long-range cis-regulatory element domain) in the latest version of the manuscript. Furthermore, to facilitate a better understanding, we prepared a new density plot of LCREDs, as shown in Fig 2C. We have corrected the corresponding figure legends as follows:

Page 36, lines 888-891: Density plot showing the size distribution of merged H3K27ac regions and long-range cis-regulatory element domains (LCREDs) of genes that were defined using the “single nearest gene” rule proposed in GREAT (<http://great.stanford.edu/public/html/>). The median values are indicated by the vertical lines.

- **"Given the close relationship between the SH-induced H3K27ac decrease and human diabetes (Supplemental Fig. 7) " How do decreased K27ac relate to human diabetes? This cannot be left to the reader to figure this out by digging into a suppl figure.**

We apologize for this inconvenience in the original manuscript. We have moved the information in Supplemental Fig 7 to a new Fig 3B, with some changes in the corresponding figure legend, as below.

Page 36, lines 905-907: Analysis of decreased H3K27ac regions. Six motifs are identified. Motifs for HNF1B, NEUROG2, FOXA1, and RFXDC2 were similar to those for HNF1A, NEUROD1, FOX:Ebox, and RFX6, respectively.

Furthermore, we have added a description in the text of the previous section to facilitate understanding of the reason why decreased H3K27ac is related to human diabetes.

Page 8, lines 173-181: Among the six detected motifs, four were highly similar to the known consensus motifs HNF1B ($p=10^{-38}$), NEUROG2 ($p=10^{-32}$), FOXA1 ($p=10^{-23}$), and RFXDC2 ($p=10^{-15}$) (Fig 3B), among which HNF1B is known to be responsible for maturity-onset diabetes of the young (MODY) type 5 (Horikawa et al. 1997). Furthermore, each motif was similar to that of HNF1A, NEUROD1, FOXA2 (FOX:Ebox), and RFX6 (Fig 3B), which have been established as the causative or susceptible genes of human MODY3 (Yamagata et al. 1996), MODY6 (Malecki et al. 1999), T2D (Gaulton et al. 2015), and neonatal diabetes mellitus (Smith et al. 2010), respectively, suggesting the possible involvement of similar pathological mechanisms of human diabetes in KK-SH mice.

In addition, we deleted the word “close” from the quoted sentence to avoid overstatement regarding the relationship with human diabetes.

Page 8, lines 186-188: Given the ~~close~~ relationship between SH-induced decrease in H3K27ac and human diabetes (Fig 3B), we performed a functional analysis using GREAT to gain insights into the various H3K27ac regions (Table S9).

- **"Sequence disruption of critical TF binding motifs was unexpectedly unprevailing in SH- induced H3K27ac decrease in KK mice". This outcome is not necessarily expected. Many genetic effects that underlie the KK diabetes prone phenotype are likely to act elsewhere, while many changes in K27ac are likely to be secondary. It is not even clear whether there is an enrichment of disruptive sites relative to expectation, although that would not necessarily be a requirement for something like this to happen.**

We thank the reviewers for the insightful comment. We understand that the outcome we observed was not necessarily expected; thus, we have deleted the word “unexpectedly” from the sentence (page 8, line 191). In relation to this correction, we deleted the word “unexpectedly” from the sentence in the abstract (page 3, line 59).

Within the regulatory elements showing an SH-induced H3K27ac decrease, we further tested the distribution of small variants in critical TF motifs. We used HOMER software to identify DNA sequences of known TF motifs in the reference C57BL/6J genome with a score above the pre-optimized detection threshold and found 17,438 FOXA2, 21,654 FOX-Ebox, 1,236 HNF1A, 3,284 HNF1B, 11,482 NEUROD1, and 15,219 RFX6 motifs. There were approximately 1.3-3.9% of the TF motifs containing small variants and statistical analysis indicated that SNVs of KK mice were significantly depleted in each of the motifs relative to the genome-wide distribution (Supplemental Table for Reviewer Only 2), with some exceptions for indels that were rarer than SNVs (Supplemental Table for Reviewer Only 3), providing another insight that the base pair substitutions caused by small variants are unlikely to play a primary role for most

decreases in H3K27ac level. However, according to the reviewer's comment, we decided to further examine the SNVs identified within the TF motifs.

As suggested by the reviewer, we predicted whether a variant would be disruptive to a TF motif by assessing biophysical models of the interaction between TFs and DNA. To this end, we constructed models on publicly available HT-SELEX datasets using a software named No Read Left Behind (NRLB; Proc Natl Acad Sci U S A. 115:E3692-E3701, 2018; PMID: 29610332; DOI: 10.1073/pnas.1714376115) (Table S12). We limited our analysis to SNVs because indels are rarer than SNVs and could cause unavoidable complexities arising from a wide range of base compositions. In relation to this new analysis, we have added the following description to the MATERIALS AND METHODS section:

Page 19, lines 489-502:

Estimation of TF-DNA binding affinity using the No Read Left Behind (NRLB) tool

To quantitatively estimate the binding affinity of various TF motifs with the associated TFs, we used an analytical tool named NRLB (Rastogi et al. 2018). We downloaded publicly available HT-SELEX datasets for TFs including HNF1A, HNF1B, LHX2 (Jolma et al. 2013; Yang. et al 2017), FOXA2, NEUROD1 (Yin et al. 2017), RFX6, and FOXM1 (Yan et al. 2021). After adapter sequence removal, followed by trimming reads to leave the central 30 base pairs as needed, model fitting was performed using parameters such as the number of rounds and binding mode, as indicated (Table S12). A fastq file of random sequences of the same length was generated (https://github.com/johanzi/fastq_generator) and used as a control when round 0 HT-SELEX data were not available. A mononucleotide model was generally used, except for the NEUROD1 HT-SELEX dataset, which was trained using a dinucleotide model. According to the authors' tutorial (<https://github.com/BussemakerLab/NRLB>), we generated an energy logo motif and subsequently scored the binding affinity of all observed DNA sequences of HOMER known motifs and their variants including various base pair substitutions.

Among the known TF motifs identified, there was a variety of binding affinities to the TF of interest. We observed up to three base pair substitutions in these motifs in KK mice, although substitutions of two or more base pairs were extremely rare. The estimation from all possible variations of the identified TF motifs revealed that the fewer base pair substitutions a TF motif accumulated, the more motifs were expected to show increases in binding affinity. Despite this, single base pair substitutions were expected to be disruptive for more than 75% of the motif alterations (Fig S7; Supplemental Figure for Reviewer Only 6). Although single base pair substitutions were found to cause affinity disruption in similar proportions for all kinds of motifs tested, the enrichment of disruptive sites relative to expectations were not observed. These results suggest that natural base pair substitutions in TF motifs are disruptive and are under natural selection in various directions, depending on the type of TF motif. Based on these results, we have added the following sentence to the text:

Page 9, lines 205-209: We found that, for more than 75% motifs including an SNV, single base pair substitution caused disruption of the binding affinity as estimated by the "No Read Left Behind (NRLB)" energy prediction algorithm (Fig S7; Table S12) (Rastogi et al. 2018) and could indeed be involved in the SH-induced local decrease in H3K27ac levels.

Although we agree with the reviewer's idea that enrichment of disruptive sites would not be the requirement for SH-induced H3K27ac changes, it would be a subject of future study to investigate the mechanisms of the variants flanking regulatory elements.

- **P9 "Since the phenotype of KK-SH mice was heritable, " unclear what the authors mean by this.**

Regarding this point, we have added more description in the text as follows:

Page 9, lines 193-197: Since SH-induced weight gain and diabetes are a set of phenotypes that are transmissible from parent to child in KK mice, it could be possible to identify the associated changes in the genomic DNA sequences relative to the reference genome of C57BL/6J mice. We first tested whether sequence variants were involved in the decrease in H3K27ac caused by nucleotide changes (s) in critical TF motifs (Fig 3B).

- **Concerning the TF motifs in strain-specific elements, are the TF mRNAs differentially expressed?**

We agree with the reviewer's comment and it is possible that differentially expressed transcription factors could be involved in the presence or absence of strain-specific H3K27ac regions. To be precise, it would be necessary to compare the expression levels of Lhx2 and Foxm1 mRNAs when strain-specific H3K27ac enrichment begins in specific tissues or cell types during development; however, such data are currently not available. For reference purposes, we tentatively performed a comparative analysis of whole transcriptome data between chow-fed 54-week-old C57BL/6J mice and 11-week-old KK-GH mice. We found that in pancreatic islets, Lhx2 was not expressed and Foxm1 was not differentially expressed (Supplemental Figure for Reviewer Only 7). However, this does not deny the possibility that differential expression of TF mRNAs causes strain-specific elements, which we consider as a research question to be investigated in the future.

- **"We found that 89.9% (5,222/5,806) of the H3K27ac regions contained at least one of the known motifs, including FOXM1 and LHX2"**
- **This type of finding suggests that the motifs have low complexity, and might not be very predictive of true binding.**
- **The authors should also spell out what these regions are.**
- **It would be important to know if there is enrichment for variants that disrupt the predicted affinity of enriched motifs.**

We thank the reviewer for the careful review of the manuscript. As our study focused on TF motifs identified using a single position weight matrix (PWM) model and did not consider true TF-binding events, we have discussed this issue as one of the limitations of this study in the DISCUSSION section.

Page 15, lines 362-364: Third, the TF motifs identified using a single position weight matrix model have low complexity and may not be very predictive of true binding.

Regarding the C57BL/6J strain-specific regulatory elements, we present the GREAT functional enrichment analysis data in Table S14. Although this included some enriched terms associated with endocrine pancreatic function and glucose utilization, the overall results allow for a relatively broad interpretation. To obtain a more specific view, we performed the GREAT analysis again by changing the filter of binomial fold enrichment (set to ≥ 2 -fold by default). With a more moderate threshold of ≥ 1.5 -fold, we found that the terms related to insulin secretion and action predominantly dominated the top 10 rankings, which was followed by pathological terms related to diabetes and obesity, including abnormal free fatty acids level, decreased circulating HDL cholesterol level, abnormal circulating lipoprotein level and abnormal fetal

growth/weight/body size (Supplemental Table for Reviewer Only 6). Based on these observations and Fig S12, we have come to understand that while the set of C57BL/6J strain-specific regulatory elements is not a complete collection of susceptibility loci for diabetes and related diseases on its own, it has the potential to form a fully connected cis-regulatory network when joined by other complementary elements.

With regard to the enrichment of the disruptive variants, we performed the same analysis for the distribution of small variants within the identified TF motifs, as described earlier in response to the reviewer's comment (Table S12; Fig S7; Supplemental Figure for Reviewer Only 6). Using the HOMER software, we found 12,337 FOXM1 and 10,906 LHX2 motifs in the C57BL/6J strain specific H3K27ac regions, among which we found that approximately 1.7-3.0% of the TF motifs harbored small variants (Supplemental Table for Reviewer Only 4 and 5). We found a significant depletion of SNVs in LHX2 motifs, although indels were significantly enriched in the FOXM1 motifs, suggesting the enrichment of disruptive indels in the FOXM1 motif.

We also predicted whether an SNV was disruptive to a TF motif using the No Read Left Behind (NRLB; Proc Natl Acad Sci U S A. 115:E3692-E3701, 2018; PMID: 29610332; DOI: 10.1073/pnas.1714376115). As observed earlier, the estimation from all possible variations of identified known FOXM1 or LHX2 motifs revealed that the fewer base pair substitutions a TF motif accumulated, the more motifs were expected to show an increase in binding affinity (Supplemental Figure for Reviewer Only 8). We also observed that single base pair substitutions were expected to be disruptive for more than 74% of the motif alterations (Fig S9; Supplemental Figure for Reviewer Only 8). In the case of observed variations in KK mice, although single base pair substitutions were found to cause affinity disruption for similar proportions of both FOXM1 and LHX2 motifs, both motifs revealed a significant underrepresentation of disruptive sites relative to expectations (Supplemental Figure for Reviewer Only 8). Therefore, these data suggest that natural base-pair substitutions in the TF motifs of FOXM1 and LHX2 are disruptive and under natural selection in various directions. Based on these results, we added the following sentences to the text:

Page 10, lines 241-245: As described above for the SH-induced H3K27ac decrease, we estimated using NRLB that for more than 74% of motifs including an SNV, single base pair substitutions were disruptive to the binding affinity (Fig S9; Table S12) and thus could cause a pre-determined local H3K27ac absence. However, in most cases of KK-selective H3K27ac absence, sequence disruption of the enriched motifs is unlikely to be a direct cause.

- **"Next, we focused on the SH-induced differences in the H3K27ac regions" What is the specific question here? SH-induced differences in the H3K27ac regions have already been examined in earlier sections.**

We thank the reviewer for highlighting these ambiguous descriptions. The specific question here was whether there are roles for sequence variants outside of H3K27ac in the H3K27ac response to SH. We have revised the text as follows:

Page 11, lines 255-257: Therefore, we investigated whether sequence variants outside of H3K27ac had any impact on the various types of H3K27ac changes in the pancreatic islets of KK mice.

Page 11, lines 263-265: Next, given the relatively small role of sequence variants inside H3K27ac in SH-induced H3K27ac changes as already mentioned, we investigated whether sequence variants outside H3K27ac had any impact on the H3K27ac response to SH.

We have made significant changes to the sentences around this part of the text, according to the next comments raised by the reviewer.

- **"Although they were complicated by various directions of change (Fig. 6A), the variant density was significantly higher in the LCREs where at least one decreased H3K27ac region was included" How many H3K27ac regions can be included in a LCRE?**
- **Which variants exactly? (KK or KK vs C57?). This analysis seems somewhat discordant with the integration of variants with SH dependent motifs. Why is this presented separately from the rest of the SH dependent analysis of variants? Perhaps this is a different type of variant analysis? This is in any case not explained.**

Our analysis revealed that up to 33 H3K27ac regions were included in a 556-kb-long LCRED that was assigned to the *Ptprn2* gene, whereas up to 66 H3K27ac regions were identified in the same LCRED using the same peak calling method for the downloaded H3K27ac ChIP-Seq data. Given that the FastQC results of our H3K27ac ChIP-Seq reads showed a significantly lower %GC and less deviation from the theoretical normal distribution than the other downloaded data (Supplemental Figure for Reviewer Only 2), we speculate that the lower maximum numbers of H3K27ac regions in our samples were due to the successful detection of short read coverage in the regulatory regions with lower GC content, resulting in moderately elongated H3K27ac regions without fragmentation into smaller pieces. We believe that this idea is supported by the observation that the average GC content of whole H3K27ac regions in the *Ptprn2*-LCRED was significantly lower in our samples than that in the others (45.7% vs. 46.6%; $p < 0.05$, two-tailed Welch's *t*-test), thus validating our experiments.

With regard to the term "variants" used in that sentence, we used it in the sense of genetic differences in KK mice relative to the mouse reference genome of C57BL/6J, which was the initial sequence firstly reported in 2002 (Nature 420:520-562, 2002; PMID: 12466850; DOI: 10.1038/nature01262) to which all subsequent sequences for other mouse strains were compared. To facilitate clarity, we have corrected this sentence as follows:

Page 11, lines 265-268: Although they were complicated by various directions of change (Fig 7A), the variant density of KK mice in the mm10 assembly setting was significantly higher in LCREDs where at least one decreased H3K27ac region was included (Fig 7B; Fig S11C; Table S6 and S7).

Regarding the analysis of variant density in LCREDs, we found it useful to incorporate the reviewer's suggestion regarding the controlled genomic regions raised at the beginning of the comments. To perform the genomic segmentation using the unsupervised learning method named ChromHMM (Nat Protoc 12:2478-2492, 2017; PMID: 29120462; DOI: 10.1038/nprot.2017.124), we defined an 18-chromatin-states model by integrating ChIP-Seq data of the 6 most fundamental kinds of histone modification as controls, including H3K4me1 (SRR9840936 and SRR9840937), H3K4me3 (SRR6728249), H3K27me3 (SRR6728238), H3K9me3 (SRR6728239), H3K36me3 (SRR6728243) and H3K27ac (SRR6728247) (Fig 5). We found that small variants in KK mice were enriched in genomic regions that lacked these modifications (Fig S10), supporting our data indicating strong enrichment of variants flanking regulatory elements. Because we thought it was inadequate to discuss such small variants in relation to the TF-binding motifs in accessible

genomic regions, we presented the data separately to develop a novel concept of variant function. As an introduction to the new analysis, we have added several sentences. We have also added a description in the MATERIALS and METHODS sections about the use of ChromHMM and data interpretation.

Page 10, lines 246-255: Since the results obtained thus far did not place much emphasis on the role of small variants in the TF-binding motifs in accessible genomic regions, we next examined how SNVs and indels of KK mice were distributed and found that the density of sequence variants was significantly higher outside than inside H3K27ac (Fig 4F; Fig. S8C). To further characterize this observation, we used ChromHMM (Ernst and Kellis 2017) for unsupervised segmentation of the genome. Using an 18-state model derived from the islet ChIP-Seq data of fundamental 6 kinds of histone modifications, including H3K4me3, H3K27me3, H3K9me3, H3K36me3, H3K27ac (Lu et al. 2018), and H3K4me1 (Wortham et al. 2023) (Fig 5A; Fig S10A), we found that both SNVs and indels were preferentially distributed in the genome segments indicating a relative paucity of these markers (Fig 5B and C).

Page 22, lines 563-597:

Characterization of chromatin states of mouse pancreatic islets using ChromHMM

We performed unbiased genome segmentation using ChromHMM (Ernst and Kellis 2017) to annotate the chromatin states in mouse pancreatic islets. Since our analysis focused on H3K27ac, we trained the extended 18-state model, instead of the ‘core’ 15-state model (Kundaje et al. 2015) without H3K27ac, by integrating ChIP-Seq data of histone modifications, including H3K4me3, H3K27me3, H3K9me3, H3K36me3, H3K27ac (Lu et al. 2018), and H3K4me1 (Wortham et al. 2023). The alignment of single- or paired-end sequencing data to reference mm10 genome was performed using Bowtie2-2.3.5.1 by the default settings, whereas an option ‘-N 1’ was added to allow a mismatch in a seed to increase sensitivity for H3K27me3 and H3K9me3 data. Following generation of binarized data using a command ‘java -mx1200M -jar ChromHMM.jar BinarizeBed -b 200’, the training of data was performed through 740 iterations using a command ‘java -mx1200M -jar ChromHMM.jar LearnModel’. Given the variations in epigenetic profiles of enhancers and bivalent and quiescent states among tissues, as reported previously (Velde et al. 2021), we provided annotations and abbreviations to individual genomic segments as follows; 1. Active TSS (TssA), 2. Flanking TSS (TssFlnk), 3. Flanking TSS Upstream (TssFlnkU); 4. Flanking TSS Downstream (TssFlnkD); 5. Strong transcription (Tx), 6. Weak transcription (TxWk), 7. Genic enhancer 1 (EnhG1), 8. Genic enhancer 2 (EnhG2), 9. Active Enhancer (EnhA) 10. Weak Enhancer 1 (EnhWk1), 11. Weak Enhancer 2 (EnhWk2), 12. Repeats (Rpts), 13. Heterochromatin (Het), 14. Bivalent/Poised TSS (TssBiv), 15. Weak Enhancer 3 (EnhWk3), 16. Weak Repressed PolyComb (ReprPCWk), 17. Quiescent/Low 1 (Quies1) and 18. Quiescent/Low 2 (Quies2). Among our annotations, the 12th segment showed enrichment both for H3K9me3 and for H3K27me3, as well as for the other active markers including H3K27ac, which may correspond to the previously deprecated cluster named as ‘Repeats (Rpts)’ because the mapping artefacts to repetitive elements was suspected (Kundaje et al. 2015). By integrating the annotations of RepeatMaster obtained from the UCSC Genome Browser (Smit AFA, Hubley R, Green P., RepeatMasker Open-3.0. <http://www.repeatmasker.org>. 1996-2010.), we found that these genomic regions were divided into two groups: one with almost 100% coverage of repeat elements and the other with almost no repeat sequences. The use of paired-end data in our analysis helped reduce the possibility of artifacts. Moreover, an example of H3K9me3 domains showing enrichment of H3K27ac, H3K4me1, and H3K36me3 has been reported (Thorn et al. 2022), implying its relevance, at least in part. We found that these regions occupied the smallest part (0.17 %) of the genome and thus had little impact on this study, and we tentatively named them Repeats (Rpts). In addition, our results did not include genomic segments corresponding to

those initially referred to ‘Bivalent Enhancer (EnhBiv)’. Based on these considerations, we analyzed the statistical differences in the densities of SNVs or indels in these genomic segments.

- **"In line with GREAT (McLean et al. 2010)" Please explain what functionalities of GREAT were used and how.**

We thank the reviewer for pointing out the incomplete explanation of the data analysis. The concept proposed by GREAT (Nat Biotechnol 28:495-501, 2010; PMID: 20436461; DOI: 10.1038/nbt.1630) was epoch-making in that it could include distal cis-regulatory elements for functional enrichment analysis by using a binomial test, whereas conventional methods could use only proximal elements by using a hypergeometric test. Because using only a binomial test can cause false-positive results arising from gene-specific enrichment of proximal and distal elements, rather than from enrichment involving multiple genes associated with a certain term, the GREAT output, by default, comprises only enriched terms significant by both binomial and hypergeometric tests. However, in our analysis, we selected all results that were significant only for the binomial test. According to the above literature, we also selected the “single nearest gene” rule from three options for the definition of LCREDs and the susceptible genomic loci for various GWAS diseases/traits, so that all the distal information about H3K27ac regions could be included in the enrichment analysis. Hypergeometric tests were not included because we were interested in covering both types of results with term- and gene-specific enrichment. Furthermore, we filtered the GWAS diseases/traits with number of associated genes still less than 5 in order to focus on more robust evidence. According to the reviewer's comment, we have made the following corrections to the text:

Page 12, lines 285-288: Motivated by the analytical strategy of GREAT (McLean et al. 2010), we utilized a binomial test to assess the enrichment of susceptible diseases/traits in the H3K27ac regions in both proximal and distal elements, the output results of which included both term-derived and gene-specific enrichment (see Materials and Methods).

Page 20, lines 507-513: We took advantage of the concept of GREAT (McLean et al. 2010) in terms of gene assignment and used only a binomial test, although the GREAT output, by default, comprised only enriched terms significant by both binomial and hypergeometric tests to avoid false-positive results arising from gene-specific enrichment of proximal and distal elements rather than from enrichment involving multiple genes associated with a certain term. We selected all results that were significant only for the binomial test, because we were interested in covering both types of results with term-originated and gene-specific enrichment.

Although we were able to obtain reasonable results using our analytical approach of assuming candidate effector genes in each GWAS locus, it should be noted that another reviewer critically expressed concerns about the uncertainty associated with this speculation. According to your constructive suggestion, we have added the following statement as one of the limitations of this study:

Page 15, lines 365-367: Fifth, since the causal effector genes at each type 2 diabetes GWAS locus have not been determined yet, we have to say that our approach of taking the disease annotation, principally based on nearest genes, is an approximation.

- **Figure 6E, figure legend. "two types of repressive epigenomic changes..." Why are these changes repressive? What are the named genes? 1 is alpha cell specific, 2 are duct**

specific, why are these highlighted in particular? According to the text 10 are "candidate genes for islet dysfunction", why? Next sentence also alludes to "repressive H3K27ac changes", these are not necessarily "repressive" events.

We agree with the reviewer's comment that the two types of epigenomic changes, including KK-selective absence and SH-induced decrease, are not necessarily repressive and have corrected the quoted sentence as follows:

Page 38, lines 953-955: Venn diagram showing the number of type 2 diabetes susceptibility genes associated with KK-selective absence and/or single-housing induced decrease in H3K27ac. Of all the 467 genes, ten that showed downregulated expression after single-housing are shown.

Regarding the differentially expressed genes listed in Fig 7E, we initially selected candidate genes for islet dysfunction and highlighted some of them with the aid of information on both cell type specificity and expression levels from single-cell RNA-Seq datasets (<https://sandberglab.se/tool/pancreas>; Cell Metab 24:593-607, 2016; PMID: 27667667; DOI: 10.1016/j.cmet.2016.08.020). However, based on the reviewer's constructive comment, we believe that such information does not provide a direct explanation for islet dysfunction. Therefore, we listed all 10 downregulated genes harboring either or both types of insufficient H3K27ac status in KK-SH mice and suggested relevant published articles for reference as follows.

Page 14, lines 352-355: Existing studies suggest that only a few out of ten candidate genes identified using pancreatic islets (Fig 7E), including *CSMD1* (Steen et al. 2013) and *DOK5* (Cai et al. 2003), are involved in the mechanisms of glucose homeostasis and insulin signaling at the molecular level, respectively, suggesting that many challenges remain to be addressed in future research.

We further made corrections to the sentences in the text where we inappropriately used the word "repressive" as follows:

Page 12, lines 300-306: Although the highest enrichment was observed for both SH-induced H3K27ac decrease and KK-selective H3K27ac absence, a large proportion (~30%) of associations with T2D loci were provided by either of the two types of insufficient H3K27ac levels (Fig 7E), suggesting an environmental role in the T2D pandemic (Ogurtsova et al. 2017; Zheng et al. 2018). Of the 467 T2D GWAS genes harboring insufficient H3K27ac, 10 genes were found to be significantly downregulated in the pancreatic islets of KK-SH mice, which showed different patterns of association with either or both types of insufficient H3K27ac levels (Fig 7E; Table S1).

- **Minor points:**
- **The KK Model, described as polygenic and inbred, should be explained more clearly in the introduction because it is not a very common strain.**

In accordance with the reviewer's constructive comment, we have added some descriptions to the Introduction as follows:

Page 4, lines 81-88: KK mice belong to an inbred strain originally derived from experimental albino mice obtained from Kasukabe district in the suburbs of Tokyo, Japan (Kondo et al. 1957; Ikeda 1994). Initially, this strain was established through a selection regime for reduced variance in body weight, gentle nature, and high reproductivity to produce an ideal experimental mouse

line (Kondo et al. 1957) but was reported to be hereditarily obese and diabetic several years later (Nakamura 1962). Although it is regarded as a model of diabetes because of insulin resistance caused by mild obesity (Dulin and Wyse 1970), defects in the insulin secretory response to glucose and pathological changes in beta cells exist in these mice (Ikeda 1994).

Page 4, lines 90-94: By the beginning of the 2000s, genetic approaches, such as QTL mapping, had identified several susceptibility loci associated with diabetes and related diseases in KK mice, suggesting their polygenic nature (Suto et al. 1998; Shike et al. 2001). Environmental factors, including diet, have been reported to play a role in the development of diabetes (Matsuo et al. 1971).

- **Fig 1b-d provide labels for the two colors.**

In Fig 1B-D, we show the time-course analysis of body weight, non-fasting blood glucose levels, and non-fasting plasma insulin levels. The data are shown using either bars or boxes in each graph for comparison of two categories: group-housed and single-housed KK mice, colored green and red, respectively. To facilitate distinction between the labels, we have changed the figure legend as follows:

Page 35, lines 858-867: (B) Body weight of KK-GH (green bars) and KK-SH (red bars) mice. KK-GH, n=11 (9- to 11-week-old) or 10 (13- to 15-week-old); KK-SH, n=11 (9- to 11-week-old) or 10 (13- to 15-week-old). (C) Non-fasting glycemia of KK-GH (green bars) and KK-SH (red bars) mice. KK-GH, n=11 (9-week-old) or 10 (11- to 15-week-old); KK-SH, n=11 (9-week-old) or 10 (11- to 15-week-old). The data for mice at 9 weeks old were obtained 2 days after grouping. n.d., not done. A linear trend test was performed to determine age-dependent increase in blood glucose levels for each group. p values were calculated using multiple t tests using the two-stage step-up false discovery rate method of Benjamini, Krieger, and Yekutieli. (D) Non-fasting insulin of KK-GH (green boxes) and KK-SH (red boxes) mice. KK-GH, n=10; KK-SH, n=10. The data for mice at 9 weeks old were obtained 2 days after grouping.

- **"11-week-old KK-SH mice showed symptoms" the authors surely mean metabolic features or metabolic changes.**

We thank the reviewer for the careful review of the manuscript. We rewrote the quoted phrase to "11-week-old KK-SH mice showed metabolic features" (page 6, line 125).

- **Why do the authors carry out de novo motif analysis if they will later discard them if they do not match expected consensus motifs?**

Although de novo motif analysis is significant because it provides unbiased identification of the most overrepresented motifs from a large number of divergent DNA sequences, it has also been pointed out that the strategy of estimating known motif enrichment can outperform it for datasets showing fewer degrees of freedom in the DNA sequences, as illustrated by the examples of PU.1 (GAGGAAGT) and ISRE (GAAACTGAAA) motifs harbored in GA-rich sequences (<http://homer.ucsd.edu/homer/introduction/basics.html>). Our dataset of SH-induced H3K27ac increases revealed preferential localization surrounding TSSs (Fig S4A), which we hypothesized to be the reason for the de novo motifs having less similarity to the canonical sequences due to the GC-rich background. Therefore, we performed a known motif analysis and found strong

enrichment of the NRF1 (CTGCGCATGCGC) motif in these elements. Regarding this point, we have added the following explanation in the text:

Page 19, lines 478-483: De novo motif enrichment for the differential H3K27ac regions was also assessed relative to the custom background sequences of KK mice using HOMER (Heinz et al. 2010) with the command “findMotifsGenome.pl” with options -size given -mask -mset vertebrates, which was also useful for the identification of the enrichment of known canonical TF motifs especially in the DNA sequences with less degree of freedom (<http://homer.ucsd.edu/homer/introduction/basics.html>).

- **"by disruption of the transcription factor (TF)-binding motif" should be plural "of TF binding motifs**

Table 1 could be reduced to 10-5 without losing useful information

We thank the reviewer for highlighting this. We changed the wording to “by disruption of the transcription factor (TF)-binding motifs” (page 4, line 71). We also corrected Tables 1 and 2, in which the top five enriched results are presented.

Responses to the comments of Reviewer #2

- **Environmental factors are known to influence the epigenomic landscape, but more detailed studies are required. In this very straight forward but compelling study, the investigators addressed the influence of living alone on type 2 diabetes risk in a murine model. The findings are very notable, and I have no concerns about how this study was conducted or interpreted. The only additional limitation I would ask the authors to list is that the community is still not entirely sure what the causal effector genes are at each type 2 diabetes GWAS locus, so taking the annotated name, principally based on nearest gene, is an approximation leveraged in this current study.**

We thank the reviewer for the helpful comments and constructive suggestions. We have substantially improved our manuscript based on these valuable comments.

We completely agree with the reviewer's comment that our approach of assuming effector genes based on the nearest genes is an approximation, and we have added the following sentence in the text:

Page 15, lines 365-367: Fifth, since the causal effector genes at each type 2 diabetes GWAS locus have not been determined yet, we have to say that our approach of taking the disease annotation, principally based on nearest genes, is an approximation.

May 20, 2024

RE: Life Science Alliance Manuscript #LSA-2023-02099-TRR

Dr. Kazuki Yasuda
National Center For Global Health and Medicine
Department of Metabolic Disorder, Diabetes Research Center, Research Institute,
1-21-1 Toyama
Shinjuku-ku, Tokyo 162-8655
Japan

Dear Dr. Yasuda,

Thank you for submitting your Research Article entitled "Single-housing induced islet epigenomic changes are related to polymorphisms in diabetic KK mice". It is a pleasure to let you know that your manuscript is now accepted for publication in Life Science Alliance. Congratulations on this interesting work.

DISTRIBUTION OF MATERIALS:

Again, congratulations on a very nice paper. I hope you found the review process to be constructive and are pleased with how the manuscript was handled editorially. We look forward to future exciting submissions from your lab.

Sincerely,
